# Predicting categorical and continuous Alzheimer's disease outcomes from a single MRI scan

Daren Ma [1] ✉, Christabelle Pabalan[2], Abhejit Rajagopal[1], Akanksha Akanksha[2], Yannet Interian[2], Yang Yang [1] ✉ & Ashish Raj [1,3] ✉

Deep learning (DL) has shown success in predicting Alzheimer's disease (AD) diagnosis, yet continuous measures such as cognitive assessment remain critical for richer prognosis, trajectory tracking and clinical trial enrichment. Current neurocognitive batteries are time-consuming, and the few DL models predicting cognition require expensive multimodal neuroimaging and longitudinal data. Although magnetic resonance imaging (MRI) is the most clinically accessible modality, on its own it struggles to capture AD heterogeneity in modern DL frameworks. We propose a multitask DL strategy integrating domain knowledge with large pretrained models to predict cognitive scores using only baseline MRI and demographics. By customizing loss functions and leveraging tissue segmentation-tuned latent representations as regularization features, our approach bypasses the need for longitudinal, multimodal or specialized neuroimaging data. This knowledge-informed multitask framework produces accurate diagnosis, segmentation and both current and future cognitive scores from a single baseline scan, with broad implications for early diagnosis, prognosis and clinical trial design.

AD is a neurodegenerative disorder affecting upwards of 50 million people and 60–70% of dementia cases worldwide. Many risk factors of its progression are known, including apolipoproteins (APOE), amyloid plaques, neurofibrillary tangles and loss of neural pathways[1–3]. There is great interest in quantifying progression using noninvasive imaging such as MRI[4,5]. The advent of modern machine learning (ML) and DL techniques based on deep neural networks (DNNs) have produced robust and accurate results for classification tasks such as early detection, diagnosis (for example, AD versus non-AD) and predicting conversion to dementia—several comprehensive reviews and meta-analyses are available[6–9]. However, neurological assessment uses additional criteria such as natural history and cognitive impairment typically assessed by a battery of neurocognitive tests delivered by a trained neuropsychologist—one of the most important of which is the

Alzheimer's Disease Assessment Scale—cognition subscale (ADAS–Cog)[10]. Fine-grained prognostic measures of cognitive status offer greater clinical utility than AI-suggested diagnosis alone. Another important use is in cohort enrichment in large clinical trials, where the ability to predict correctly rapid progressors from nonprogressors reduces sample sizes and hence cost[11].

Disease severity and progression rates are associated closely with cognitive score[12]; hence, an accurate prediction of cognition will also indirectly aid diagnosis and prognosis, which however remains challenging as cognition is an emergent property linked causally in complex ways to brain circuits[13] and the underlying mechanisms remain unclear[14]. A 'remarkable heterogeneity' was noted in the link between brain atrophy and cognitive scores in patients with AD[15]. Nonetheless, aided by the remarkable effect that neurodegeneration has on morphology,

[1]Department of Radiology and Biomedical Imaging, University of California San Francisco, San Francisco, CA, USA. [2]Data Institute, University of San Francisco, San Francsico, CA, USA. [3]Bakar Computational Health Sciences Institute, University of California San Francisco, San Francisco, USA. ✉e-mail: daren.ma@ucsf.edu; yang.yang4@ucsf.edu; ashish.raj@ucsf.edu

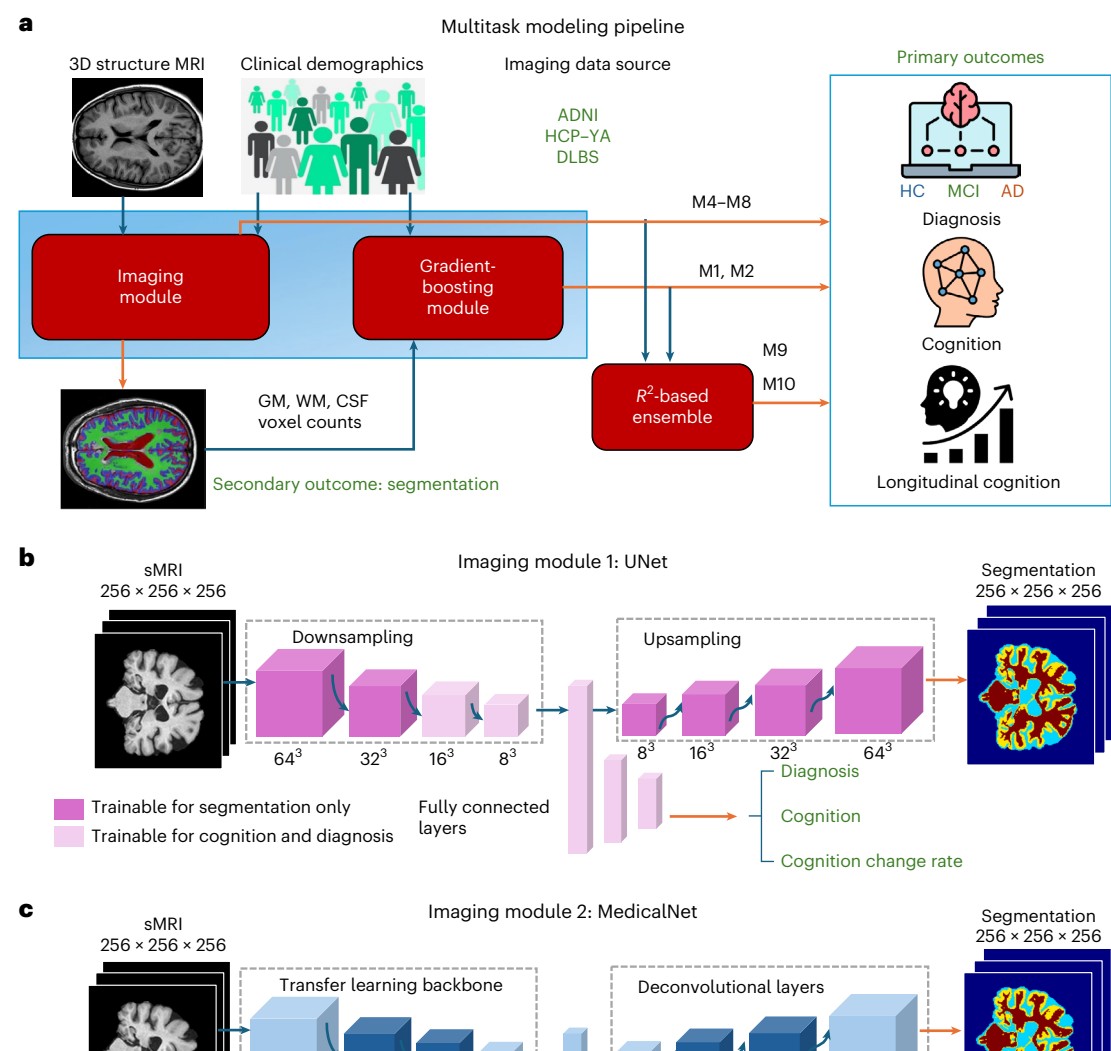

**Fig. 1 | Overall model architecture. a**, Flowchart showing the overall experiment design. Imaging module takes the baseline 3D sMRI as input and performs segmentation, then feeds the post-training low-dimensional latent variables to fully connected layers for primary outcomes predictions, including diagnosis, cognition and longitudinal changes. The nonimaging tree-based gradient-boosting module takes the participants' clinical demographics and the tissue-class segmented volumes as input, and predicts the multitask outputs. An $R^2$ ensemble is applied to combine both modules for better results. The model numbers on the arrows represent the corresponding models in Table 2. Blue arrows: model input; orange arrows: model output. **b**, Detailed structure of the custom UNet model, which is trained on the tissue segmentation task. The features from its most concentrated convolutional block are extracted to predict scaler targets. After flattening the 128 channels of 4 × 4 × 4 matrices into a feature vector of length 8,192, three fully connected layers are then fine-tuned to generate the multihead outputs. Middle layers of the UNet contracting arm and the fully connected layers are trainable for the primary targets, as denoted by lighter shades. **c**, Tencent's MedicalNet structure is applied as our second approach for the imaging module. We freeze the middle layers in both the model backbone and deconvolutional (upsampling) layers and fine-tune other model weights using the specific task of predicting scalar outputs from baseline sMRI scans.

prediction of cognitive decline in AD from MRI should be theoretically possible. Cognitive hotspots such as the hippocampus become smaller with age, and structural brain changes are associated with more rapid cognitive decline[14–16] and higher rates of conversion to dementia[17,18].

Unfortunately, modern AI tools have been less successful in predicting continuous measures such as cognition in AD cohorts using neuroimaging[19], which not only impacts clinical utility but also leaves open a critical gap of some of the most important questions in human neuroscience[20]: the neural correlates of brain–mind connection, or how image-level morphological features govern downstream high-level cognition. We identify three key challenges that limit translational impact:

(1) Successful classification does not yield usable continuous measures of cognitive impairment. Transfer learning on a pretrained ResNet50 model could predict conversion to dementia robustly but correlated poorly to cognitive decline[21]. Grand Challenges involving hundreds of participating laboratories have not produced good predictions of cognitive performance, using genetic, cognitive, positron emission tomography (PET) and MRI data[7,22,23]. The DREAM challenge yielded modest prediction of cognitive decline from a diverse set of models, with no clinical value in using genetic or imaging data[22,24]. The TADPOLE challenge, using several biomarkers gave modest prediction of

**Table 1 | Demographic distribution of the main dataset used for training and CV (ADNI), the pretraining dataset for segmentation (HCP–YA) and the independent dataset for out-of-sample testing (DLBS)**

| Characteristic | Training set | Validation set | Testing set |
|---|---|---|---|
| ADNI (*N*=1,950) | | | |
| Gender (F; M) | 738; 822 | 97; 98 | 93; 102 |
| Diagnosis (CN; MCI; AD) | 380; 848; 332 | 56; 89; 50 | 18; 132; 45 |
| Marriage (married; divorced; other) | 1,171; 137; 252 | 151; 17; 27 | 142; 25; 28 |
| Age, years | 73.1 (50.4–91.4) | 72.9 (55.9–90.1) | 72.7 (55–90.0) |
| Years of education | 16.1 (4–20) | 15.9 (6–20) | 15.9 (7–20) |
| MMSE | 26.8 (3–30) | 27.1 (6–30) | 26.9 (0–30) |
| ADAS–Cog11 | 10.2 (0.7–38) | 9.6 (0.7–37) | 10.4 (1.0–42.7) |
| HCP–YA (*N*=1,008) | | | |
| Gender (F; M) | 554; 466 | | |
| Diagnosis (CN; MCI; AD) | 1,008; 0; 0 | | |
| Age, years | 28.3 (22–37) | | |
| DLBS (*N*=331) | | | |
| Gender (F; M) | | | 213; 118 |
| Diagnosis (CN; MCI; AD) | | | 331; 0; 0 |
| Age, years | | | 62.5 (25–93) |
| Years of education | | | 15.9 (11–21) |
| MMSE | | | 29.0 (25–30) |
| ADAS–Cog | | | 5.2 (0–19.3) |

Mean values (range) over all participants is shown, with a typical separation of training, validation and set-aside testing sets, reflecting random sampling followed by splitting into three groups in the ratio 8:1:1. The HCP-YA and DLBS datasets are not split as all of the participants are cognitive normal (CN) controls. The out-of-sample testing DLBS participants are all healthy, younger than ADNI participants and generally have better cognitive scores.

ADAS–Cog13 scores, which were more difficult to forecast than clinical diagnosis or ventricle volume[7,23]; even the best-ranked model was unable to outperform informed random guessing or simple mixed-effects regression.

(2) Meta-analytical studies report that MRI has little incremental value for classification tasks in comparison with nonimaging clinical and cognitive biomarkers[9,25]. It was reported to be the least predictive amongst several modalities[8,9], and was not supported for early diagnosis of dementia as a stand-alone add-on test[26]. DNNs give substantially better classifier performance when several expensive neuroimaging modalities (MRI and PET), several visits and clinical, cognitive and fluid biomarkers are used together. Very few instances report good classifier accuracy using a single MRI scan alone, without PET, cognitive or clinical biomarkers[6,9,25,27].

(3) Historically, DNNs have not performed well using raw MRI input compared to those that extract regional features following time-consuming volumetric or morphometric image processing pipelines such as Freesurfer[28] or ANTS[29]. For example, even the modest cognition predictions in TADPOLE required well-selected regional morphometric features instead of convolutional neural networks (CNNs) trained directly on MRI[6,23]. This reliance on time-consuming preprocessing and previous feature selection can affect translational utility and imposes model-dependent decisions[19].

These challenges necessitate a rethink of how off-the-shelf AI technologies are deployed and scaled in neurodegenerative disease research—this is the purpose of the current study. Our objective was to achieve simultaneously the following goals: (1) segment the brain MRI into gray matter (GM), white matter (WM) and cerebrospinal fluid (CSF) tissue classes; (2) predict diagnosis; (3) predict continuous measures of disease severity—specifically, the patient's current and future cognitive scores. We wish to achieve these goals with only a single three-dimensional (3D) MRI scan, no longitudinal data, no PET or other modalities, no genetic or fluid biomarkers and no cognitive assessments at baseline. The scope of these goals was driven by an overriding requirement for clinical translational relevance in real-world scenarios, especially community clinics, where typically only structural MRI may be available. However, it is important to note that our approach is not a substitute for other biomarkers and modalities already reported in the vast AD literature. On the contrary, successful AI-based integration of MRI is meant to enable further enhancements that include additional biomarkers when available.

A static MRI scan might appear unlikely to relate adequately to high-level cognition and its longitudinal decline. As repeated previous artificial intelligence (AI) studies find, and our analysis confirms, a conventional single-shot CNN tasked with predicting ADAS–Cog is hampered by insufficiently large samples and by the extreme dimensionality mismatch between voxel-level imaging input and scalar cognitive output.

Instead we present a careful curation of AI strategies, with two contrasting but nonexclusive approaches. In the first approach, we used a 3DResNet architecture, widely popular in image labeling[30], and further retrained on medical imaging data[31]. We refer to this as the 'MedicalNet' model in the rest of the paper. We retrain the MedicalNet judiciously on AD-specific data through transfer and adaptation learning to predict cognition. This approach leverages knowledge distillation[32], domain adaptation[33] and transfer learning[34–36]. The second approach relies heavily on domain knowledge, custom loss functions and custom input features, and leverages the excellent soft-tissue contrast of MRI to capturing low-level tissue segmentation and supply reliable low-dimensional predictive features. To this end we train a 3D UNet module to achieve 3D segmentation of the MRI into GM, WM and CSF classes, and use its bottom layer to capture the implicitly conditioned neuroimaging features for cognition-sensitive structure. The segmentation task serves as a regularization tool in a data-scarce setting, and its fast runtime serves, in certain use-cases, as an alternative to dedicated advanced and computationally expensive neuroimaging pipelines such as FreeSurfer[28]. Both domain knowledge-driven and transfer learning strategies are essential, and together they are more successful than black-box DL.

The resulting multitask framework (Fig. 1) is combined by means of an ensemble model to yield highly effective outcomes, equaling or exceeding state-of-the-art in all three tasks up to 36 months out. Surprisingly, the custom 'small' UNet trained from scratch gives comparable performance to retrained large ResNet50 model from MedicalNet, highlighting the critical value of proposed customizations. We implemented nine separate benchmark ML models (M1–M9) and on alternative input features; none produced comparable performance. Finally, we performed out-of-sample studies to demonstrate generalizability. These results paint a compelling picture supporting proposed framework's performance and broad applicability. We are not aware of any previous study that has attempted these goals with only baseline MRI and minimal demographics. The present study has deep translational implications in early diagnosis, prognosis and clinical trial design.

## Results

This study uses three public datasets for training, cross-validation (CV) or out-of-sample testing; demographics and data processing details are in Table 1 and Supplementary Methods 1. Using baseline demographic and MRI, we wish to achieve three tasks at baseline (tissue segmentation, diagnostic classification, cognition prediction) and also longitudinal cognition. Presented models achieve all three tasks simultaneously, but cognition prediction was the dominant target. We used a mix of models: XGB that ingests tabular data, and three image models that

**Table 2 | Main task modeling results**

| No. | Model | Input | | Segmentation | Cognition | | | Diagnosis |
|---|---|---|---|---|---|---|---|---|
| | | demographics | MRI | Dice (CV; test) | Loss | CV $R^2$ (mean ±s.d.) | Test $R^2$; MAE | Accuracy (CV; test) |
| M1 | XGB (standard MSE) | Yes | No | –; – | 86.19 | 0.26 ± 0.067 | 0.24; 6.69 | –; – |
| M2 | XGB (gamma loss) | Yes | No | –; – | 72.3 | 0.31 ± 0.047 | 0.27; 6.45 | –; – |
| M3 | Single-task CNN | No | ADNI | –; – | 64.64 | 0.45 ± 0.029 | 0.42; 4.18 | –; – |
| M4 | Multitask UNet | No | ADNI | 0.9410; 0.9266 | 58.92 | 0.62 ± 0.035 | 0.60; 4.92 | 92.76%; 88.75% |
| M5 | UNet | No | ADNI +HCP | 0.9479; 0.9361 | 52.05 | 0.68 ± 0.072 | 0.66; 4.31 | 92.54%; 88.72% |
| M6 | MedicalNet | No | ADNI +HCP | 0.9556; 0.9325 | 58.77 | 0.61 ± 0.068 | 0.58; 4.48 | 91.34%; 90.26% |
| M7 | UNet | Yes | ADNI +HCP | 0.9582; 0.9388 | 54.40 | 0.72 ± 0.053 | 0.68; 3.97 | 91.79%; 90.77% |
| M8 | MedicalNet | Yes | ADNI +HCP | 0.9691; 0.9639 | 46.91 | 0.74 ± 0.059 | 0.70; 2.82 | 92.14%; 91.28% |
| M9 | **Ensemble (UNet+XGB)** | **Yes** | **ADNI +HCP** | **0.9813; 0.9740** | **37.96** | **0.83 ± 0.050** | **0.82; 2.48** | **94.76%; 92.82%** |
| M10 | **Ensemble (MedicalNet+XGB)** | **Yes** | **ADNI +HCP** | **0.9795; 0.9654** | **40.88** | **0.83 ± 0.045** | **0.80; 2.29** | **93.79%; 92.30%** |

After evaluating an initial set of models (M1–M4), we focused on the UNet and MedicalNet models (rows 5–10) with matching model settings between them. The last two rows represent the ensemble models, highlighted by boldface, show the results from our two best performing proposed multitask ensemble models that achieved field-leading performance for all three tasks, with segmentation Dice scores above 0.96, $R^2$ of cognition task above 0.8 and diagnosis accuracy above 92% in the set-aside testing set. CV: ninefold CV result; Test: performance on set-aside test set.

ingest baseline MRI: CNN, MedicalNet and UNet. These components are illustrated in Fig. 1. Throughout, CV, out-of-sample testing and rigorous assessment of algorithmic choices in an incremental fashion was performed, as summarized in Table 2 and Supplementary Table 2.

### Prediction of baseline outcomes

**Segmentation.** Dice scores of segmentation performance after nine-fold CV and out-of-sample testing is listed in Table 2. The DNN models were able to generate segmentation results visually indistinguishable from the 'silver standard' provided by FSL–FAST[37] (Fig. 2a). The multi-task UNet (M5) achieved validation Dice of 0.998, 0.964, 0.951 and 0.973 for background, GM, WM and CSF, respectively. The UNet+XGB model (M9) with custom Gamma loss achieved overall Dice of 0.9740, whereas MedicalNet+XGB (M10) achieved Dice of 0.965 (Fig. 2b). These numbers compare favorably with most cutting-edge segmentation algorithms ('Discussion'). Notably, best performance required use of demographic data via XGB, suggesting demographics contain some relevant tissue information, and its inclusion induces a desirable regularization effect on model training and generalization.

**Diagnosis.** Table 2 and Fig. 2c,d summarize the accuracy of various models on the diagnosis task (AD versus non-AD). The UNet, when trained exclusively with structural MRI (sMRI) inputs alone, achieved a testing diagnostic accuracy ($n = 195$) of 88.72%, whereas MedicalNet achieved 90.26%. Testing accuracy increased to 92.82% and 92.30%, respectively, when we adapted with demographic and segmentation information and ensemble approach (Fig. 2d). Confusion matrices for the diagnosis task for all benchmark models are contained in Supplementary Table 7. Receiver operating characteristic (ROC) curves (Fig. 2c) show that ensemble models (M9, M10) achieved over 0.9 area under the ROC curve (AUC). Thus, ensembling and the incorporation of diverse data modalities enhances performance. Also, the diagnosis accuracy for all benchmark models on the cross-validated sets ($n = 1,755$) are reported in Supplementary Table 8, on the set-aside testing set ($n = 195$) are reported in Supplementary Table 9. Supplementary Fig. 4 illustrates further details of the confusion matrices and ROC curves for the best models in three subsets: overall ($n = 1,950$), CV ($n = 1,755$) and set-aside testing ($n = 195$).

**Prediction of baseline cognition.** Table 2 summarizes performance on the baseline ADAS11 cognition task of all models, reporting the CV and set-aside testing results on the Alzheimer's Disease Neuroimaging Initiative (ADNI) cohort. The XGB regressor (M1) with only clinical demographics produced an $R^2$ of around 0.24. Implementing Gamma loss on XGB (M2) yielded slightly better results. The single-task CNN

model (M3) could improve $R^2$ to only 0.42. A multitask UNet trained on ADNI MRI solely (M4) achieved accurate segmentation, diagnosis and cognitive prediction, but inclusion of the Human Connectome Project (HCP) and use of segmentation volumes in training improved $R^2$ to 0.66 (M5) and 0.58 (M6). Adding demographics to the input layer of the UNet (M7) and MedicalNet (M8) improved $R^2$ to 0.68 and 0.70, respectively.

*Ensemble models.* The $R^2$-weighted Multitask UNet+XGB (M9) showed the best $R^2$ CV range of 0.78–0.87 and testing $R^2$ of 0.82, mean absolute error (MAE) of 2.48. Ensemble MedicalNet+XGB (M10) reached $R^2$ CV range of 0.77–0.87 and testing $R^2$ of 0.80, MAE of 2.29 (Fig. 2f). From the respective $R^2$ values of the cross-validated performance of the two models being ensembled, M2 and M7 (Table 2), we observed that the ensemble approach weighed the XGB portion by ~30%, and the UNet portion by ~70%. Notably, these models were not trained separately on disease cohorts, yet all groups were predicted successfully. Supplementary Table 3 shows that although prediction accuracy is consistently high and comparable across groups, as expected it is lowest for mild cognitive impairment (MCI) participants, arguably the most clinically heterogeneous group.

**Joint prediction of all phenotypes.** Overall, models jointly incorporating demographics, MRI and segmentation tend to show improved prediction accuracy in all tasks (Table 2). Our best ensemble multi-task UNet with XGBoost (M9) produced a testing segmentation Dice Score of 0.9740, testing cognition $R^2$ of 0.80, coupled with the best set-aside testing accuracy of 92.82%. Similarly, Tencent's MedicalNet with XGBoost (M10), using demographics and segmentation gave a set-aside testing segmentation Dice Score of 0.9654, highest testing cognition $R^2$ of 0.82 and testing diagnosis accuracy of 92.30%. These numbers hit the upper limit of most previous benchmarks models trained on single tasks and indicate the essential value of proposed customizations. In Supplementary Fig. 1 we show that prediction of cognition becomes more accurate by adding the diagnosis task.

**Imparting additional rigor and assessing model choices.** We report several strategies (Supplementary Sections 3–7) to enhance rigor and assess the effect of individual model choices:

- Use of our custom Gamma loss over conventional MSE loss significantly improved cognitive prediction performance ($\Delta R^2$ up to 0.05; Supplementary Table 1).
- Introduction of segmented GM, WM and CSF volumes as DNN input further increased $R^2$ (by 0.05–0.08; Supplementary Table 1).

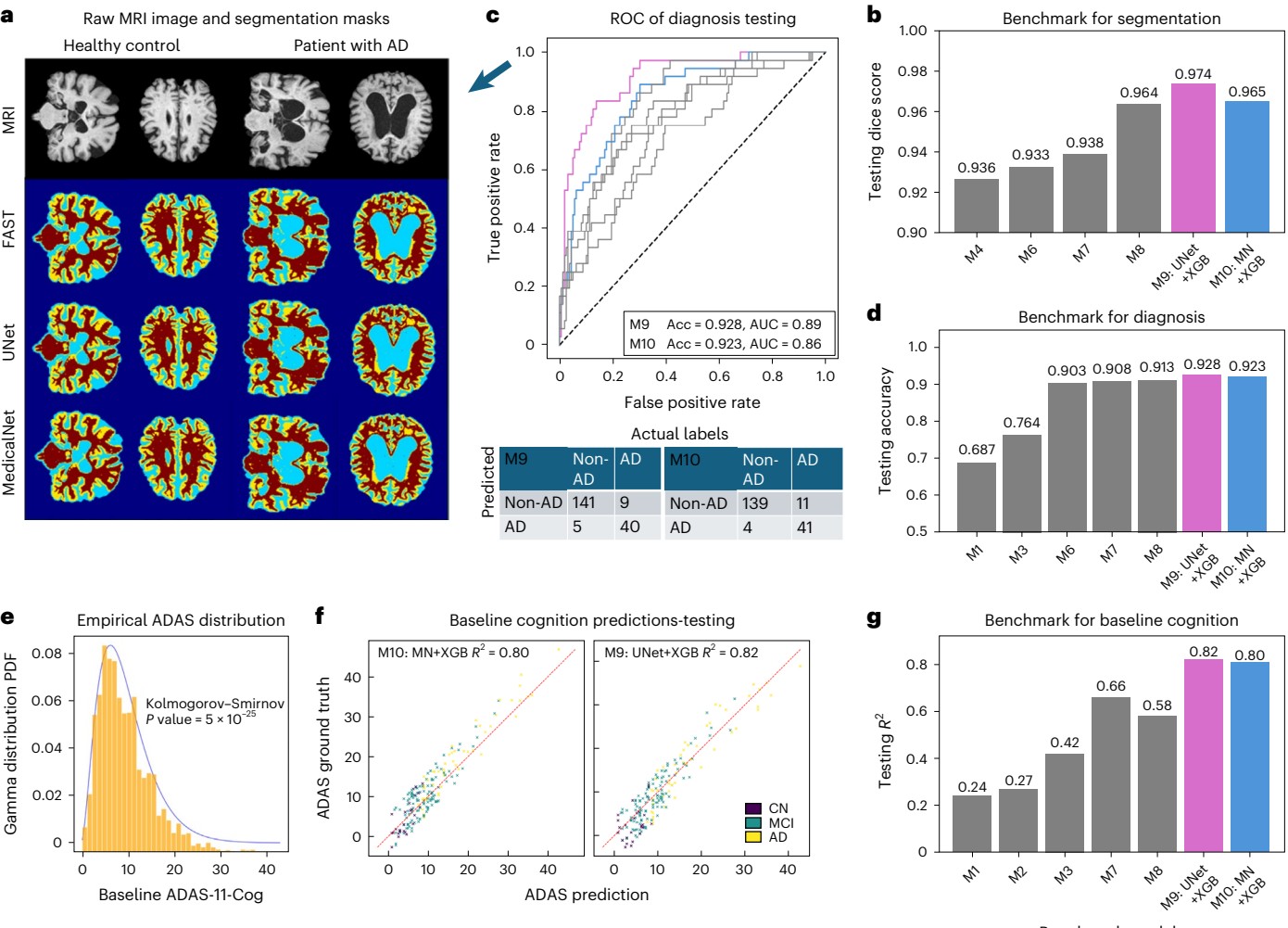

**Fig. 2 | Baseline performance of multitask models: set-aside testing.**
**a**, Coronal and axial slices of two representative participants demonstrating segmentation outcomes. The FSL–FAST segmentation results were used as the training target, and our best models (multitask UNet and multitask MedicalNet) both yielded strong performance on the segmentation task. **b**, Barchart of segmentation Dice scores of all benchmark models, highlighting the superiority of UNet and MedicalNet compared with other benchmark models (M4, M6, M7, M8) that did not involve the ensemble approach. Model orders correspond to Table 2. **c**, ROC curve highlighting the two best models on diagnosis task on the set-aside testing dataset. Gray AUC curves represent benchmarks from M4 to M8. The confusion matrix below shows that the UNet model is more robust against class imbalance. Acc: accuracy. **d**, Accuracy of the diagnosis task, including M1, M3, M6, M7 and M8 and the two best models, on the set-aside testing dataset of 195 participants. Refer to Supplementary Fig. 4 for the diagnosis results on the full dataset. **e**, Baseline ADAS–Cog11 scores from ADNI follow a Gamma distribution, with a two-sided Kolmogorov–Smirnov test distance of 0.04 and $P$ value $< 10^{-5}$. **f**, Primary outcome of baseline ADAS–Cog11 predictions from the best performing UNet and MedicalNet models, reaching $R^2 > 0.8$ in the set-aside testing set. **g**, The testing $R^2$ of baseline cognition prediction task for benchmark models M1, M2, M3, M7 and M8, highlighting that the best performance is achieved by the ensemble UNet M9 and MedicalNet M10.

- Randomized CV: repeating tenfold CV 20 times with randomized participant orders closely mirrored the data in Table 2 (Supplementary Table 1).
- Model selection: XGB was shown superior to other conventional ML models (Supplementary Table 2), whereas 3DResNet50 was superior to the smaller 3DResNet10, both from the MedicalNet codebase.
- Hyperparameter tuning: was achieved through a grid search for the best hyperparameters of the DNNs (Supplementary Table 4 and Supplementary Fig. 2).

## Prediction of longitudinal cognition

Switching now to the clinically more interesting but challenging problem of predicting future cognition from baseline demographics and MRI, Fig. 3a plots empirical versus predicted ADAS–Cog11 change rate, measured by exponential coefficient $\alpha$ (see 'Loss functions of the model architecture' in Methods), from both ensemble MedicalNet and UNet. For the longitudinal portion, we filtered out MCI or AD participants whose longitudinal ADAS–Cog showed improving cognition, which we ascribe to operator noise. Figure 3b reports the predictions of cognitive scores segregated by month of visit (12- and 24-month) of CV, where ground truth cognitive score at baseline is known; future cognition is computed using fitted $\alpha$ from above and the known baseline cognition. Clearly, the predictions of ADAS–Cog11 are highly accurate for up to 24 months. Figure 3c depicts the situation when the ground truth cognition at baseline is not known; now future cognition is estimated using fitted $\alpha$ and the predicted baseline cognition from Fig. 2. Even with unknown baseline, our ensemble models achieved Pearson's $R > 0.7$ and $R^2 \approx 0.6$ at 24 months. Figure 3d shows how the model performance varied by visit month. The abnormal decrease and increase at months 18 and 30 may be due to comparatively lower sample size at these visits. Overall, both UNet and MedicalNet are robust and accurate up to 30 months after the initial visit. Even at 36 months, the models continue to be significantly predictive, with $R^2 \approx 0.5, P < 10^{-6}$.

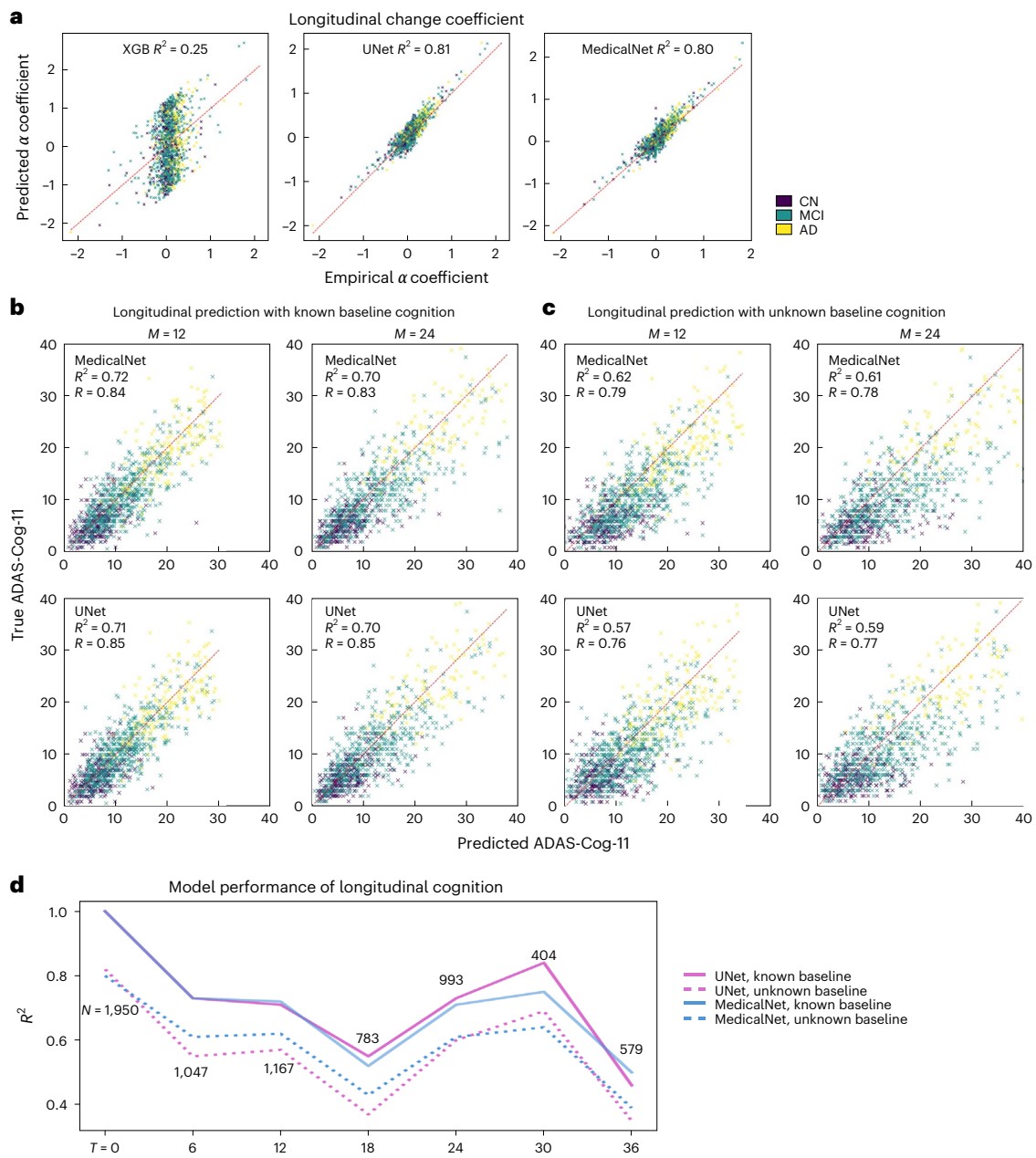

**Fig. 3 | Prediction of future cognition from baseline data. a**, Scatterplots of performance of XGB and the ensemble UNet and MedicalNet models in predicting Longitudinal Change Coefficient $\alpha$. **b**, Scatterplots of empirical versus predicted longitudinal ADAS−Cog-11 at 12 months and 24 months from each participant's initial scan, used as input by each model. Predicted cognition was computed using $\alpha$ fitted above, and the known baseline cognition of the participant. Note the high and highly significant $R^2$ and Pearson's $R$, displayed within each panel. **c**, Longitudinal predictions using unknown baseline cognition. This case required the longitudinal prediction to be based on both the fitted $\alpha$

from **a** as well as predicted baseline cognition from Fig. 2. The resulting $R$ and $R^2$ numbers are lower compared to **b**, yet still very high. **d**, Plots of model performance of longitudinal predictions over time, from baseline visit to 36 months. The sample size available at each visit is indicated. The general trend shows that the predictive power of the models goes down gradually, yet there is a performance drop at 18 months and a bump at 30 months, probably caused by the smaller sample size at those visits. Even at 36 months, the models continue to be significantly predictive $R^2 \approx 0.5$. The $P$ values for these one-sided $R^2$ statistics showed $P < 10^{-4}$ for all models at 18 months, and $P < 10^{-6}$ everywhere else.

These regression scores are some of the highest reported in the literature ('Discussion').

Overall longitudinal prediction trajectories as a function of time are shown in Supplementary Fig. 3. For models M9 and M10, longitudinal predictions of the whole trajectory reported over 0.8 on Pearson's $R$.

**Feature importance and visualizations**
Figure 4a highlights the relative importance of demographics and tissue volume features (violins) and of sMRI (bars); the latter indicate

significantly higher importance than nonimaging features. Among the scalar features, age, marriage status and GM volume are the most important components for predicting baseline cognition in both M9 and M10, whereas age and years of education are the most important for predicting future decline using the change coefficient $\alpha$.

Next we visualized the importance of each image element to the cognition predicted by the UNet, using the mean occlusion map[38] over all patients with AD (Fig. 4b). The posterior parietal, medial and lateral temporal, and frontal cortices appear as hotspots, aligning closely with

established neuroanatomical correlates of cognitive impairment in AD. Remarkably, however, the occlusion hotspots are dispersed widely in a multifocal fashion, and do not seem to occur uniformly within gross gyral boundaries. A region-wise illustration of occlusion maps in Fig. 4c shows that temporal and medial limbic cortical areas exhibit the highest scores, along with bilateral hippocampi and amygdalae, consistent with their known role in cognitive function and AD pathology (Supplementary Section 9 and Supplementary Table 5). We further uncover the internal model elements contributing to the cognition task with a receptive field analysis we reported previously[39]; Methods. Figure 4d shows the local receptive field feature representation of our UNet, depicting extraction of intermediate feature manifolds. Colors correspond to pseudolabels, which contribute differently to regression scores. Three participants' channel slices are displayed to help visualize the neural network 'trace' of the input query image. The descending arm captures image-level features that concentrate the most important features for the ADAS prediction; these features correspond roughly to tissue classes.

### External out-of-sample testing

We performed out-of-sample testing on the external Dallas Lifespan Brain study (DLBS) dataset for the pretrained multitask UNet to predict the ADAS–Cog scores (Fig. 5a), which yielded an $R^2$ score of 0.63, emphasizing the generalizability of our proposed approach. Figure 5b depicts a representative participant's segmentation output. The average Dice score over the 331 DLBS participants was 0.923. In addition to the testing performance on the best model, we tested the DLBS data on all models (Supplementary Table 6). These results indicated that there is a performance drop between ADNI testing and DLBS testing. We attribute the performance difference primarily to domain shift, as the datasets differ in scanner distribution, demographic composition and acquisition parameters. For instance, the ADAS–Cog11 scores in DLBS range from 0 to 19.3, with an average of 5.2, yet the average ADAS–Cog11 of ADNI cohort is near 10, and the distribution is 0–42.7 (Table 1).

### Alternate models trained on Freesurfer volumetrics

The results of two benchmark XGB models trained on Freesurfer volumetric data are shown in Fig. 5c–e. The first model, trained on seven broad volumetric features, achieved a mean cross-validated $R^2$ of 0.42, and MAE of 3.63. (Fig. 5c). The second, trained on all 86 parcellated DK regions, achieved a higher cross-validated $R^2$ of 0.53, and a lower MAE of 3.46 (Fig. 5d). These results compare unfavorably to the results from our best models trained directly on T1-MRI scans (Fig. 5e). Thus, the use of these time-consuming and specialized computational pipelines did not yield competitive performance against our approach.

## Discussion

We developed a multitask ensemble DNN to simultaneously perform image segmentation, predict diagnosis and predict current and future cognitive scores. The latter remains underaddressed by AI/ML, with few successes compared to the easier task of diagnosis. To ensure clinical adoption, only a single MRI scan and common demographic information was required. Previous DL applications in neuroimaging have been criticized for overfitting, poor CV, limited sample sizes, severe dimensionality mismatch and lack of biological interpretability[19]. To overcome these challenges, we used a carefully adjudicated mixture of strategies: one of the largest available sample sizes; transfer learning from large pretrained models; ensemble methods; domain knowledge-based custom loss functions; and shared representations and regularization induced by the adjacent segmentation task.

### Summary of key findings

We trained our models on two large public datasets (ADNI, HCP Young Adult (HCP–YA)), involving 1,950 and 1,008 participants, respectively—more than most comparable studies on cognitive impairment. By means of comprehensive CV, we demonstrated simultaneous prediction of all three tasks, accomplished to a best-in-class level of accuracy. In particular, our prediction of cognitive scores comfortably outperformed other tested models and previous reports (see 'Previous AI and ML approaches using imaging data'). A key finding is that the strongest results require the most domain knowledge-driven customizations, whereas performance suffered dramatically in their absence. Surprisingly, we found that the custom 3D UNet trained from scratch gives comparable performance to the retrained foundation MedicalNet model that has twice as many trainable parameters, highlighting the critical role of these customizations. To predict future levels of cognition we re-used the tissue-wise voxel counts calculated from segmentation—a fitting example of biologically motivated data reuse—and reported accurate predictions of future cognition up to 36 months out, using only baseline imaging and demographics.

In contrast to many ML models lacking interpretability, we performed thorough feature importance analysis, which revealed sMRI as the most important contributor to cognition, followed by age, marriage and education. Occlusion analysis highlighted hotspots of cognition in the dementia brain, especially lateral (or medial) temporal, posterior parietal, cingulate, hippocampus and amygdala. These hotspots are multifocal and do not follow gross gyral boundaries. That cognitive hotspots broadly reflect the topography of AD pathology and canonical neural correlates of cognitive function, strongly supports our predictive model, and may inform future explorations of brain–behavior associations in other conditions.

We also implemented nine alternative predictive models encompassing both DNNs and conventional ML; none compared favorably to the proposed model. Similarly, secondary-image-derived features such as tissue volumes and Freesurfer parcellations were less predictive than the proposed sMRI-based framework, despite their established predictive value. Finally, an independent, out-of-sample study (DLBS) demonstrated the generalizability of our approach without further retraining. These data and benchmarks together impart compelling support to our approach.

---

**Fig. 4 | Feature importance visualizations for the cognition task.**
**a**, Permutation importance of tabular features (demographics and segmented tissue volumes) are shown as distributions over all participants in violin plots, computed using the permutation method on the XGB model. Age, marriage status and GM volume are the most important for predicting baseline cognition (left), whereas age and years of education are the most important for predicting future decline using the change coefficient $\alpha$ (right). For comparison, the relative inclusion importance of sMRI as input into the imaging module is also shown alongside (bars), indicating that, as a whole, imaging data are far more significantly important than nonimaging data. V_: volume. **b**, Occlusion importance map of imaging data showing the relative importance of every voxel in the 3D MRI in terms of its contribution to the cognition task, measured by increase in MSE at baseline, plotted on a surface map. **c**, Regional surface visualizations of occlusion maps showing lateral and medial views of occlusion scores averaged over voxels within individual ROIs. The most prominent ROIs are: IP: inferior parietal; MTG: middle temporal gyrus; Fusi: fusiform; Ling: lingual; Am: amygdala and HP: hippocampus. **d**, Local receptive field analysis applied to present a feature representation of our 3D UNet, depicting extraction of intermediate feature manifolds with corresponding pseudolabels (principal components through clustering) indicated by color. Here we selected three participants' channel slices to display the voxel-level information to help visualize the neural network 'trace' of the input query image. The descending arm captures image-level features that correspond roughly to tissue classes. For instance, L4 shows the last layer before the regression block; it concentrates the most important features for the ADAS prediction. The ascending arm captures somewhat different features than the descending arm.

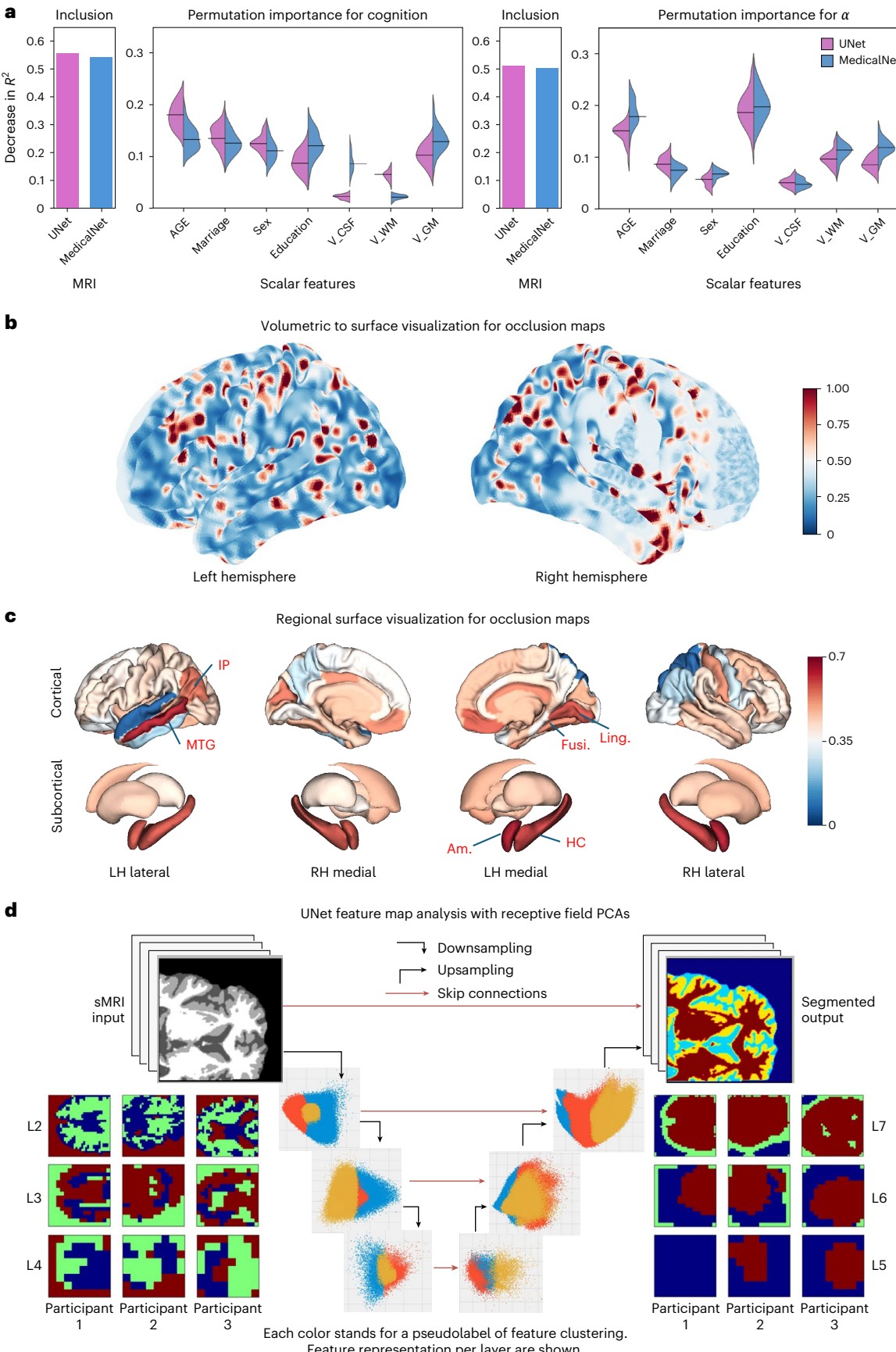

**a** Inclusion · Permutation importance for cognition · Inclusion · Permutation importance for α

**b** Volumetric to surface visualization for occlusion maps

Left hemisphere · Right hemisphere

**c** Regional surface visualization for occlusion maps

Cortical · Subcortical

LH lateral · RH medial · LH medial · RH lateral

**d** UNet feature map analysis with receptive field PCAs

sMRI input · Downsampling · Upsampling · Skip connections · Segmented output

Each color stands for a pseudolabel of feature clustering.
Feature representation per layer are shown.

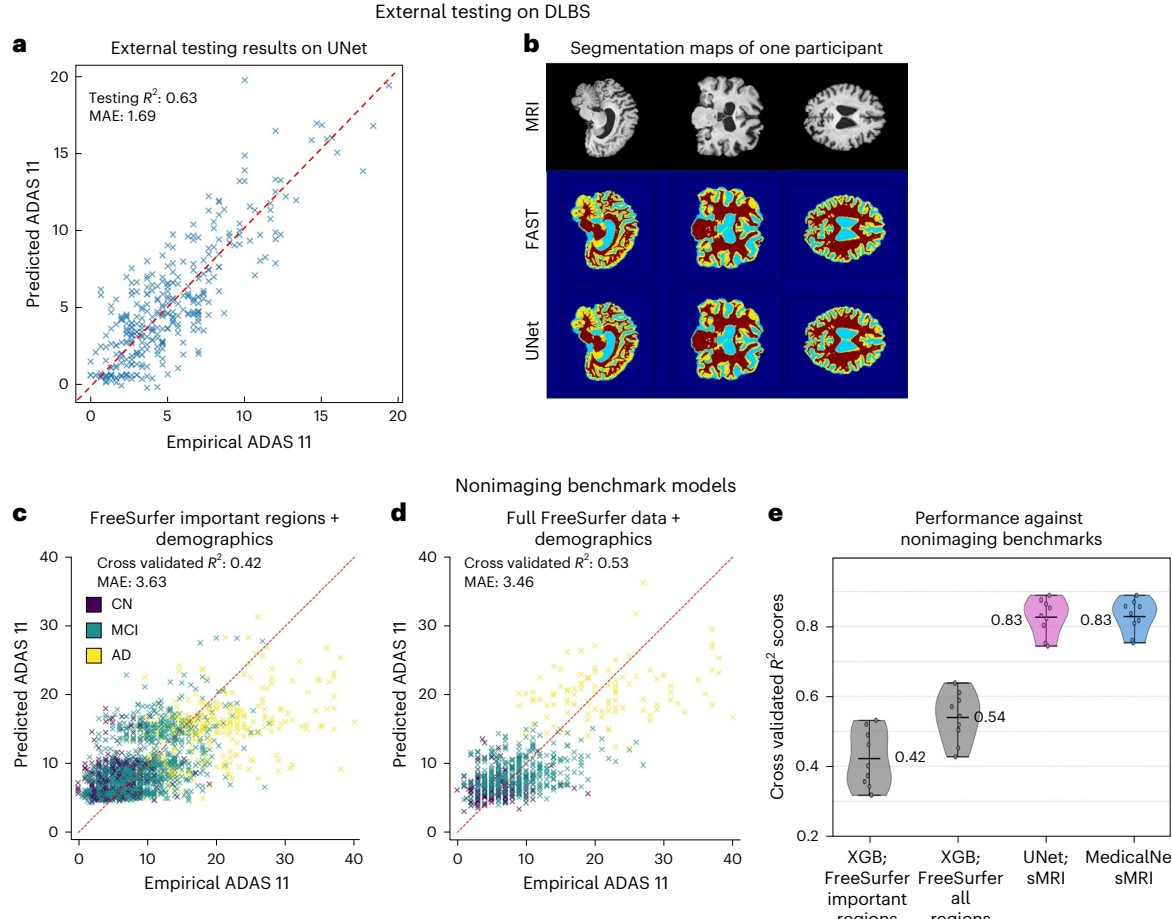

**Fig. 5 | Performance on external out-of-sample test dataset and on volumetric inputs. a**, Scatterplot comparing our multitask UNet's predicted ADAS scores against empirical ADAS scores on the external testing DLBS data (*N* = 331). With no additional training, the UNet trained on the main ADNI study was able to achieve high accuracy in this independent sample, with $R^2$ = 0.63 and MAE = 1.69. **b**, The multitask UNet was also able to accurately segment the independent dataset's MRI scans; a representative participant's example is shown. The average testing Dice score across all DLBS participants was 0.923. **c**, Scatterplot of the cross-validated predicted versus empirical ADAS scores, using FreeSurfer-derived broad tissue or regional volumes and demographics.

This model achieved $R^2$ = 0.42 and MAE = 3.63. **d**, Similar scatterplot, but now for an XGB model also trained on all 86 Desikan–Killiany regions' FreeSurfer volumes. This model achieved $R^2$ = 0.53 and MAE = 3.43. The FreeSurfer meta ROIs in **c** and **d** (and important regions in **e**) include ventricels, hippocampus, wholebrain, entorhinal, fusiform, middle temporal gyrus and intracranial volume. **e**, Violin plot comparing these volumetric-based models with our proposed multitask models (M9 and M10) that ingest MRI scans directly; we note that the latter greatly outperform the former. The red text showed the mean $R^2$ values of CV of each group, and the top and bottom bars display the highest and lowest $R^2$ scores within each group.

## Previous AI and ML approaches using imaging data

We highlight the relationship of the present approach with key studies from existing literature that have used imaging data to predict various Alzheimer-related outcomes.

**AI and ML for diagnosis.** AI, ML and DNNs have predicted effectively AD diagnosis[40,41] and conversion to dementia[41–45] from baseline imaging. A very large ResNet model trained on MRI gave classification accuracy of 91.3% (ref. 46). A multimodality CNN trained on 2,861 MRI scans achieved an AUC of 92.01% (ref. 47). 3D CNN architectures using additional baseline modalities such as hippocampal shape features from MRI[48], functional MRI (fMRI) features[49,50] or PET[51–53] and extensive demographic and clinical data, neurocognitive batteries and APOE status[54], can predict conversion status. In general, however, MRI alone is considered the least predictive among several modalities[6], has little incremental value over clinical and cognitive biomarkers for classification, and is not supported for early diagnosis of dementia as a stand-alone test[8,9,25,26].

**AI and ML to predict cognition.** Due to the remarkably complex and heterogeneous relationship between cognition and brain circuits[13–15],

previous grand challenges such as TADPOLE and DREAM have not produced good predictions of cognitive performance until now, even when using multimodal genetic, cognitive, PET and MRI data, and report no clinical value in cognition prediction using MRI alone[7,22–24]. MRI-derived volumetry changes over time perform much better than baseline MRI[55,56]. A meta-analysis of MRI biomarkers for predicting cognition[19] indicates prediction accuracy ranges from *R* = 0.2 to 0.8, with a mean around 0.5.

Sparse kernel methods applied to morphometric MRI features[57] reported *R* = 0.57 for predicting baseline ADAS–Cog. Modern ML models trained on volumetric and nonimaging features of BIOFINDER and ADNI cohorts reported successful diagnosis but weak prediction of 4-year mini-mental state examination (MMSE) slope ($R^2$ = 0.175 on test set, $R^2$ = 0.044 out-of-sample), whereas a DNN trained on raw MRI showed even worse performance[27]. Support vector regression reported *R* = 0.61 for predicting baseline and *R* = 0.46 for future change in ADAS–Cog[58]. When including PET and CSF data, the results improved to *R* = 0.74 and *R* = 0.53, respectively. The comparatively small sample sizes (*n* = 586 and *n* = 186, respectively) warrants replication of both these studies on larger samples. Multikernel regression on MRI and PET

regional features produced $R^2 = 0.53$ in predicting delayed recall—a different measure related to cognition. However, when only MRI features were used, $R^2 = 0.36$ was achieved[59]. These methods typically did not adopt current standards of sample size, CV and out-of-sample testing.

Recent DNN methods have predicted robustly conversion to dementia, but not cognitive decline[21], suggesting that successful classification does not yield usable continuous measures of cognition. A recent DL model reports $R = 0.6$ for predicting change in clinical dementia rating (CDR) score[60]; however it included only patients with AD and used preprocessed MRI morphometrics and clinical and cognitive measurements as inputs. Choi et al.[61] applied CNN on amyloid- and fluorodeoxyglucose–PET images to predict conversion to dementia; their model output showed good correlation with 3-year ADAS11 change ($R = 0.24$). Performance of our approach exceeds these reports comfortably, despite using far fewer features or visits.

**Scientific versus predictive value of AD biomarkers.** In recent years, numerous studies have examined the utility of various biomarkers such as hippocampal shape features from MRI[48], fMRI features[49,50], fluid biomarkers, neurocognitive batteries and APOE status[54], and PET[51–53,62–64]) for predicting subsequent cognitive decline across the broad spectrum of AD. The predictive value of biomarkers in these studies is typically assessed statistically, focusing on identifying the most clinical prognostic features. Tau–PET in particular is highly specific and sensitive biomarker of dementia conversion[62–64], although less so for predicting cognitive scores. We discussed above the utility of many such biomarkers in ML, yet their value extends beyond into understanding which types of biomarker (for example, blood or CSF protein levels, specific brain regions such as hippocampus) are more contributive than others. Insight into these issues can have profound implications for mechanistic insights and therapeutic interventions (for example, by stimulating such regions through neuromodulation). Hence, the present contribution should be evaluated narrowly in terms of predictive and utilitarian value in a clinical context, rather than the broader landscape of AD biomarkers.

**Transfer learning.** As off-the-shelf DL implementations have not yielded sufficiently accurate cognition predictions[7,27], we experimented with various means of leveraging large, pretrained models—here the ResNet50[65] architecture implemented in Tencent's MedicalNet[31]. A previous transfer learning model was successful at classification but less so in predicting cognitive scores[21]. The power of transfer learning and retraining of such large models was exemplified recently by Lu et al.[46], who retrained a base Inception-ResNet-V2 model designed for sex classification for classification of AD versus healthy in the ADNI database, achieving 90.9% accuracy. Although these numbers are comparable to previous ML studies and our own classification results, it is doubtful this model will work on the cognition task. Our surprising result that the custom 3D UNet trained from scratch gives comparable performance to the retrained large MedicalNet model highlights the critical value of proposed domain-specific customizations: custom Gamma loss and use of shared representations learned from tissue segmentation. Given that our custom UNet involves half as many trainable parameters as MedicalNet, the slight performance degradation of the former may represent a favorable trade-off in medical settings, where training samples are necessarily limited, conventional means of data augmentation potentially inapplicable and privacy considerations paramount.

**Strong translational and clinical potential**

Predicting a patient's current and future cognitive score from a single MRI scan can assist neurologists in diagnostic and prognostic decision-making. Unlike previous approaches, our model does not require baseline cognitive assessment, specialized image pipelines, expensive PET scans, genetic analysis or fluid proteomics, making it a fast, accurate and easily implementable tool for most clinical settings. This circumvents the need to use highly specialized, time-consuming and computationally demanding MRI morphometry software, although these well-established, open-source and extensively documented tools will remain vital for researchers. Our technique provides other users, especially clinicians, the opportunity to benefit from implicit spatial brain representations learned by our proposed models, without requiring expertise in these computational pipelines. We reported meaningful gains in speed and performance over these pipelines, which could prove valuable in specific contexts, for example, in developing a quick clinical prediction of cognitive impairment before referring the patient to a more advanced imaging laboratory and/or full neuroradiology report. Clinical utility in practice will depend on specific use-cases and environments, and will require careful assessment in future studies.

This study may help better characterize the link between morphology and cognition in diverse scenarios beyond AD, for example, in other neurodegenerative diseases such as Parkinson's, amyotrophic lateral sclerosis and Huntington's disease. Even though forecasting future cognition is more important, accurate prediction of baseline cognition too may be helpful in community settings where the lack of specialized neurocognitive assessment skills is a huge hindrance. A related use case was noted by Stonnington et al.[57]: predicting performance on global cognitive screening tests from MRI help to distinguish delirium from dementia in patients presenting to an emergency department with confusion and no records reflecting previous mental status.

Enhanced prediction accuracy of future cognitive scores reported can aid in computational and personalized staging, prognosis and early detection. It may also have potential as a tool for cohort enrichment and progression tracking in large clinical trials of disease-modifying drugs, as cognition is the most important clinical endpoint of trials. The ability to predict progressors from nonprogressors correctly using only baseline data can reduce sample sizes, and hence cost, dramatically[11]. Future studies that include additional measurements in those cases where they are available can further improve clinical utility and aid the prediction of cognition. These include longitudinal MRI and PET, genetics[66], blood and CSF protein biomarkers[67]. A future model could learn and combine the temporal and regional associations between all available biomarkers from multimodal imaging could enable complete computational tracking of disease progression.

## Methods

### Datasets used in this study

**Main dataset for training and CV.** Data for training, testing and validation were obtained from the ADNI database (http://adni.loni.usc.edu); the data included: demographics, MRI, diagnosis and cognitive assessment.

**Pretraining segmentation dataset.** We incorporated data from the HCP–YA cohort during the pretraining phase to enhance the diversity and size of our training sample for the segmentation-only task. The inclusion of these adult brain scans, which typically exhibit minimal or no atrophy[68], helped mitigate overfitting by exposing the model to a broader anatomical variance. This strategy improved the generalizability and robustness of the segmentation models, and reduced susceptibility to segmentation errors in downstream tasks.

**Independent dataset for external out-of-sample validation.** In addition, we collected an external testing dataset from the DLBS from the OpenNeuro Platform[69,70]. This study contains multimodal data on aging participants, including 465 older participants with MRI, of which 331 participants have ADAS–Cog scores recorded in wave 2. These data are used in this study as an independent cohort to assess the generalizability of the proposed AI models trained on the ADNI cohort. As the Dallas cohort does not include mature patients with AD, it is a useful dataset on which to assess the applicability of our models without

further retraining, and to explore whether cognition prediction can succeed in participants without dementia. We obtained the T1-MRI scans of these participants, fed them into our trained models without further changes, predicted ADAS–Cog11 of these participants and assessed how well the predicted scores matched the corresponding cognitive scores contained in the DLBS database.

As in other multiphase studies, ADNI is known to show some phase-wise variability. In this study, we included baseline scans of healthy participants from the ADNI-1, ADNI-2, ADNI-Go and ADNI-3 cohorts. All scans were acquired using an MPRAGE sequence with parameters of repetition time = 2,300 or 2,400–3,000 ms, echo time ≈ 3 ms, inversion time = 900–1,000 ms and flip angle = 8°–9°. Images were obtained across several sites using scanners from three different manufacturers (GE, Siemens and Philips). The spatial resolution of ADNI-1 images was 0.9375 mm × 0.9375 mm × 1.2 mm, whereas that of ADNI-2 and Go images was 1 mm × 1 mm × 1.2 mm (ref. [71]). For ADNI-3 cohort, the participants were acquired at 3T across several vendors (GE, Siemens and Philips) using harmonized protocols, with T1-weighted MPRAGE volumes obtained at a resolution of $1 \times 1 \times 1$ mm$^3$ (ref. [72]). We did not attempt harmonization between phases for ADNI.

The ADNI study procedures were approved by the institutional review boards of all participating centers as detailed at https://adni.loni.usc.edu/wp-content/uploads/ADNI%20Acknowledgement%20List_Feb2025_clean.pdf. Written, informed consent was obtained from all participants in the study according to the Declaration of Helsinki, and the study was approved by the institutional review board at each participating site. The scientific study protocol of the HCP–YA dataset is approved by the Washington University–University of Minnesota Consortium of the Human Connectome Project (WU–Minn HCP). Written, informed consent was obtained from all participants in the HCP study. The DLBS study procedures were approved by the institutional review boards of all participating centers as detailed at https://openneuro.org/datasets/ds004856/versions/1.2.0. The study was approved by the institutional review board at each participating site.

Please refer to Table 1 and Supplementary Section 1 for more detailed information about the datasets.

## Inputs to the model

The following features extracted from the aforementioned datasets were used as input to the ensemble model architecture: (1) baseline T1-weighted anatomic MRI scans preprocessed and segmented into three tissues using the FAST tool[73] and (2) selected demographics (age, sex, years of education and marriage status). For the categorical variables (marital status and gender) we applied one-hot encoding. Our tabular data was then assembled using these demographic variables. Intermediate model results are also fed to the regression part of the models, including the segmented volumes (voxel count) of segmentation maps and the latent variables.

## Exclusions.

We purposefully omitted regional biomarker features such as atrophy, volumetrics, tau and amyloid PET standardized uptake value ratios—these require advanced processing pipelines or non-standard-of-care PET imaging, and are not readily available in community clinical settings. We did not include any cognitive information or diagnosis at baseline when training the model, as cognition and diagnosis at baseline are our outcome variables.

## Primary target 1: cognitive scores (baseline).

The first and most important outcome is the baseline ADAS–Cog11 score. First introduced by Rosen and colleagues[74], the ADAS–Cog is accepted widely as a gold standard for assessing the efficacy of anti-dementia treatments. ADAS–Cog scores are collected through cognition task-based surveys, which are time-consuming, operator-sensitive and burdensome for patients with dementia. Although other clinical measures such as the MMSE or global CDR scores are more common, they are not thought to be true cognitive measures, and display unhelpful bunching at the high end of the scale[75,76]. CDR, in particular, is limited by its length of administration, reliance on clinical judgment and collateral source information, and relative insensitivity as a measure of change in interventional studies[77]. Here we use a summary value, referred to as ADAS–Cog11, of 11 important subparts of the test[12,78]. We chose ADAS–Cog11 as our primary research goal because it is predictive of the participants' current cognitive status as well as risk of future dementia development[10,79,80].

## Primary target 2: cognitive scores (longitudinal).

To achieve the clinically important objective of forecasting future cognitive scores from baseline MRI, we built an extended model mirroring the above architecture, with two principal differences:

(1) Instead of baseline cognition, the outcome measure now is the exponential change rate parameter $\alpha$, computed for each participant by fitting an exponential curve starting from baseline cognition, that is, $ADAS_T = ADAS_{baseline}e^{\alpha T}$

(2) We added a new input feature from the segmentation map produced by the UNet: $z$-score of the total volume of each segmented tissue class compared to healthy controls. This new input feature is intended to exploit the well-known dependence of cognitive decline on baseline atrophy[14,16,18].

Although it is theoretically possible to add the longitudinal coefficient outcome variable $\alpha$ in the same multitask architecture as the above model that predicts only baseline outcomes, a separate model is more appropriate here as it needs to be trained on a different, smaller, sample size, while leveraging segmentation output that was not necessary for the baseline model.

## Primary target 3: diagnosis.

Most clinical applications of ML on neuroimaging data focus almost exclusively on the goal of predicting the (baseline) diagnosis of the participant. For the current study this is not the main goal, yet one that is readily attainable, as we expect considerable concordance between diagnosis and cognition. We chose for this purpose the ability to separate the AD-dementia group from non-AD dementia groups (healthy, MCI). The labels AD or MCI for the participants included in this study were taken directly from the ADNI database; they are based on syndromic categorization, and only the AD group may be considered to have AD-dementia. Although it is possible to achieve three-way classification into HC, MCI and AD, it is notably less successful in the field; and the ability to correctly separate progressive MCI from stable MCI remains an open challenge[81] that will require further future effort.

## Secondary target: tissue segmentation map.

Our secondary target is brain tissue segmentation from baseline MRI. There is no ground truth of tissue segmentation in ADNI; hence, we settle for a silver standard obtained from an existing software pipeline called FSL–FAST[37], a computational tool that segments a 3D image of the brain into four tissue types (WM, GM, CSF and nonbrain), while also correcting for spatial intensity variations. It is robust and reliable compared to most finite mixture model methods, which are sensitive to noise[73].

## CV and set-aside data splits

For this study we collected 1,950 ADNI participants in total, and performed a train-test split at a ratio of 9:1. This 10% testing set ($n = 195$) is set-aside and never used in the training or validating processes, until we report the testing performance of each model. On the remaining 90% of samples ($n = 1,755$), we performed ninefold CV for all the models. It is worth pointing out that, when we report the main results or decide model selections, we always refer to the validation sets in the ninefold CV process, and never relied on the training scores. To eliminate potential data leakage, that is, the possibility that the DL model 'guesses' the outcomes from patient-specific data instead of learning the true, broad underlying patterns, we made sure that the patient ID information was

preserved across all stages of data cleaning in the curated dataset, but then dropped before any model fitting, which guaranteed no overlapping participants in the splitting. Further, we applied the same random seed in performing the train-validation splitting, so that the CV results were reliable and comparable across different models.

### Data augmentation using image preprocessing

Even ADNI—one of the largest standardized imaging datasets of AD—is small in comparison to typical successes of DL, due to the privacy, costs and technical challenges in medical imaging. Therefore, we used data augmentation techniques, which are the common approaches to make the models more robust[82], to artificially generate new training images by applying transformations to existing images during the training steps alone; these included an elastic transformation to introduce shape variation to the image volumes, and a standard normalization step. The elastic transformation increases generalization ability. These topology-preserving deformations are considered generic and biologically plausible[83]. For the validation and set-aside testing sets, we performed a standard normalization, so that only their true, unaltered MRI scans are used. Details can be checked in Supplementary Section 2.

### Proposed multitask network

Because existing ML and DL methods are shown to be suboptimal for the current task, we developed a bespoke multitask network that seeks to overcome the twin challenges noted previously: small training sample size and huge dimensionality mismatch. The key idea is that the process of training a model for image segmentation task will produce low-dimensional features that will also be responsive to cognition. The segmentation task will be accomplished using the UNet or MedicalNet architecture, whereas cognition will be predicted by a regression block that also incorporates demographic tabular data. The overall multitask architecture is illustrated in Fig. 1. Its components are described below.

### Image model architecture 1: UNet

The UNet is a type of convolutional autoencoder that alleviates the bottleneck requirement by providing additional connections across downsampled and upsampled feature representations, which improves segmentation accuracy[84], especially for medical imaging tasks with limited data and the presence of heavy data augmentations[85]. The architecture consists of a contracting path where the input size is downsampled but the depth of the feature map increases, and an expanding path, where the feature map is upsampled, which helps regain that precise localization necessary for accurate segmentation. Between the two arms, skip connections are added, where contextual information in the corresponding cropped feature map from the contracting path is concatenated to the expansion layers.

Of key relevance for the cognition task, the lowest-dimensional layer of the UNet ('fully-encoded layer') is considered to contain contextual information integrated across large brain areas—this is precisely the feature-rich domain that will be expected to carry cognitive information.

The proposed UNet architecture is illustrated in the middle row of Fig. 1. The contracting and expanding paths each have four convolution steps. In the contracting path, each layer contains two $3 \times 3 \times 3$ convolutions (with a padding of 1), followed by a rectified linear unit (ReLu) and then a $2 \times 2 \times 2$ maximum-pooling operation (with strides of two in each dimension). In the expanding path, every layer consists of a $2 \times 2 \times 2$ upconvolution (with strides of two in each dimension), followed by two $3 \times 3 \times 3$ convolutions and a ReLu operation. The last layer consists of a $1 \times 1 \times 1$ convolution that reduces the number of output channels to the number of labels, that is, three labels for our segmentation task.

### Image model architecture 2: MedicalNet

MedicalNet[31] is a project by Tencent that offers a collection of pre-trained 3DResNet models alongside corresponding transfer learning

modules, aiming to improve medical image analysis. Recognizing the important impact of training data volume on DL outcomes, MedicalNet aggregates datasets across various modalities, target organs and pathologies to construct sizable datasets. In the design of Med3D network, as the goal is to learn universal feature representations by training a segmentation network, the common encode–decode segmentation architecture was adopted, where the encoder can be any basic structures that allowed the network to retrain on our 3D brain MRI data. In their original paper, the models outperformed several benchmarks in various tasks including Lung Segmentation and Pulmonary Nodule Classification, among others.

We applied 3DResNet50 and 3DResNet10 networks provided in their GitHub repository (https://github.com/Tencent/MedicalNet) to conduct the transfer learning on Brain MRI dataset with randomly initiated weights. The datasets from ADNI and HCP–YA were applied to the segmentation retraining, and ADNI was then used for cognition tasks. We fixed the predetermined weights in MedicalNet other than the first encoder and last decoder layer, and attached a fully connected module to learn task specific weights. We introduced the same gamma loss for cognition predictions. The 3DResNet50 was selected after 24 epochs of training on each, for the reason that it yielded higher segmentation Dice coefficient between the two choices. We optimized the model parameters with Adam starting from the 0.001 learning rate.

### Nonimage model architecture: XGB

We wish to enhance the power of our model by adding demographic predictors that are very low-dimensional compared to imaging data— this causes huge dimensionality mismatch whereby the tabular features can easily get lost in translation during training of deep nets. Therefore, after experimenting with various linear and nonlinear ML models, we selected a gradient-boosting regressor from the XGB library[86]. XGB shows best-tested performance and great stability against overfitting.

### Loss functions of the model architecture

**Segmentation loss.** We use combined objective for training composed of categorical cross-entropy loss and a Dice coefficient metric.

The loss function of the UNet segmentation is the cross entropy computed by a pixel-wise soft-max over the final feature map. There are four categories in the ground truth segmentation map (GM, WM, CSF and background (nonbrain)). The loss is therefore

$$L_{\text{Seg}} = - \sum_{i=1}^{n=4} t_i \log(p_i)$$

where $t_i$ stands for the true voxel value and $p_i$ is the predicted voxel value. Furthermore, we calculated the weighted Dice coefficient as our metric for evaluation:

$$D_{(p,t)} = \frac{2 \sum_l \alpha_l \sum_i \min\left(p_l^i, t_l^i\right)}{\sum_l \alpha_l \sum_i \left(p_l^i + t_l^i\right)}$$

Here $p_l^i$ and $t_l^i$ are the set label probability vectors for all voxels for the ground truth and the prediction. The vector $\alpha_l$ denotes the weight of each category.

**Diagnosis loss.** For the binary log loss of our Diagnosis task, we implemented the following loss function:

$$L_{\text{Diag}} = -\frac{1}{N} \sum_{i=1}^{N} \left[ y_i \log(p_i) + (1 - y_i) \log(1 - p_i) \right]$$

where $N$ is the number of observations in the dataset, $y_i$ is the true label of the $i$th observation separating from AD and non-AD and $p_i$ is the model's predicted probability for the $i$th observation being of the positive class, hereby AD.

**Cognition loss.** Let $y_{true}$ be the actual cognitive score that we wish to estimate, and $y_{emp}$ the empirical ADAS−Cog11 measurement from the ADNI dataset. This gives the relationship $y_{emp} = y_{true} + \epsilon$, where we introduce independent and identically distributed white Gaussian noise $\epsilon$ with variance $\sigma$ into the measurement, such that $P(\epsilon) = \mathcal{N}(0, \sigma^2)$. Under this assumption the cognition loss function is simply the conventional mean square error (MSE) loss:

$$L_{Cog}^{MSE}(y) = \frac{\sum_{i=1}^{N}(y_{emp} - y)^2}{N} \qquad (1)$$

Similarly, we define the MSE loss function for the longitudinal exponential coefficient

$$L_{\alpha}(\alpha) = \frac{\sum_{i=1}^{N}(\alpha_{emp} - \alpha)^2}{N} \qquad (2)$$

Note, the above MSE losses represent the likelihood function, and do not incorporate any previous information about the data. Hence we propose here a new loss function specifically for ADAS−Cog11, as its distribution can be approximated readily by its empirical histogram. Indeed, as shown in Fig. 2e, the raw empirical histogram is fitted closely by the Gamma distribution of order $k = 1$. Hence we assert $P(y) \propto \frac{y}{\gamma}e^{-y/\gamma}$, parametrized by a single scale parameter $\gamma$ that is estimated from the empirical distribution. Using the Bayesian posterior:

$$P(y|y_{emp}) = P(y_{emp}|y) \times P(y) = \frac{1}{\sigma^2}e^{-\frac{(y-y_{emp})^2}{2\sigma^2}} \times \frac{y}{\gamma}e^{-y/\gamma}$$

we finally obtain the custom gamma loss for the baseline cognition prediction task as per:

$$L_{Cog}^{\Gamma}(y) = -\log(P(y|y_{emp})) = K + \frac{(y - y_{emp})^2}{2\sigma^2} - \log\left(\frac{y}{\gamma}\right) + \frac{y}{\gamma} \qquad (3)$$

for some constant $K$ that may be safely ignored. At the actual model training phase, for each CV split, the parameters $\sigma$ and $\gamma$ were then calculated only within the corresponding training cohort to prevent any data leakage.

We trained the nonimaging XGB module using the tabular features alone on both the original MSE loss or the custom gamma loss (the specific choice was indicated in results tables next to each model number or name):

$$L_{NIm} = L_{XGB} = L_{Cog}^{MSE} \text{ or } L_{Cog}^{\Gamma}$$

*Multitask total loss.* Armed with the above loss functions for each prediction task, we then train a full DNN imaging model, whether UNet or MedicalNet, on the combined loss integrating the segmentation, cognition and diagnosis losses:

$$L_{Im} = \lambda_{Cog} \times L_{Cog} + \lambda_{Seg} \times L_{Seg} + \lambda_{\alpha} \times L_{\alpha} + \lambda_{Diag} \times L_{Diag} \qquad (4)$$

Here, $L_{Cog}$ stands for the cognition task loss, and can be either $L_{Cog}^{MSE}$ or $L_{Cog}^{\Gamma}$. The specific choice of which cognition loss function was used is indicated in results tables next to each model number or name. Weight parameters $\lambda_{Cog}$ and $\lambda_{Seg}$ are the corresponding weight of the loss functions for baseline cognition and segmentation, which are set at 0.5 to balance the importance of these tasks. We set the relative importance of the tasks $\lambda_{\alpha}$ and $\lambda_{Diag}$ to be both 0.25.

## Ensemble approach

To enhance the model's prediction performance on the primary targets, we applied an $R^2$-based ensemble approach between the image models and nonimage XGB models. To achieve this, an XGB module was trained on either the original MSE loss or the custom Gamma Loss. For each primary task listed in Fig. 1, we first collected the model results after training both the imaging and the nonimaging (XGB) models in parallel, then calculated the $R^2$ metric during the CV process. Using the resulting $R^2$ values as weights, we then formulated the ensemble loss:

$$L_{ensemble} = R_{Im}^2 \times L_{Im} + R_{NIm}^2 \times L_{NIm} \qquad (5)$$

This ensemble loss was finally used to retrain the entire model, consisting of both the imaging and the XGB modules, as indicated in Fig. 1. It is worth pointing out that the ensemble loss is computed for all of the baseline cognition, diagnosis and longitudinal cognition change rate tasks, with the only difference being in the specific loss functions used. Note also that the segmentation task had no ensemble because the imaging module handled it independently. If the modules had similar $R^2$, the resulting loss function would be directed to treat both modules equally during the full retraining phase. If one or the other module has exceptionally high $R^2$, then that module will be favored dominantly in the ensemble approach. This is a designed behavior as we rely on the $R^2$ scores alone to decide the relative weights without excessive tweaking. Although we are aware that other more sophisticated ensemble techniques are available, here we were focused on the simplest possible strategy, which also helps in reducing the risk of overfitting.

After the previous pretraining on segmentations, we trained all primary tasks of the model simultaneously. The full model was therefore retrained end-to-end. However, as indicated in the figure, certain layers of certain modules, specifically, some selected UNet and MedicalNet layers, did not need to be retrained; those layers are shown in darker shades in the figure. Most prominently, these 'frozen' layers include the expanding arm of the UNet or MedicalNet, which are relevant only for the segmentation task, and not for the diagnosis of cognition tasks.

## Feature importance analysis

**Permutation and inclusion importance.** As shown in Fig. 4a, we implemented the inclusion feature importance for the MRI data, whereas a permutation feature importance for the scalar variables was introduced in the XGB model. The calculation of the inclusion scores are quite simple: we computed the difference in the overall $R^2$ values for the best ensemble models and the best XGB models, which quantifies the 'boost' in model performance when we include MRI images for training. This score was the difference in $R^2$ performance between with and without imaging module−a metric intended to capture the importance of MRI inputs to the cognition task.

The permutation feature importance scores were conducted by running the ensemble model based on a random-selected train-validation split, with different resampling of one scalar feature each time, for 25 iterations. Then, we calculated the decrease in $R^2$ scores between the actual models and those with permutation, and present them as the relative importance of the certain features. The permutation part was done in Python using Numpy's resampling methods.

**Occlusion map of the UNet model.** Occlusion map is a common way to diagnose the contribution of each image section (voxel) to the prediction target of imaging models[87]. Occlusion map blocks different parts of the input image through masks iteratively and attempts to find the parts that increase the MSE loss of the cognition prediction task. We implemented a widely used occlusion map computation[38] in Python, generating a mean occlusion map across all participants with AD (Fig. 4b). This mean occlusion map was calculated by generating individual participants' occlusion maps first on a batch size of five voxels, normalizing across the participants at a global maximum value of ten, and then averaging them over all participants in the coregistered standard MNI space. As the MRI data we have worked on so far are in the voxel-level space, we used a volumetric-to-mesh projection function to

produce the surface visualizations, particularly achieved through the nifti-to-mesh function on MNI152 space in the Neuromaps toolbox[88]. The Nilearn code base[89] was used to plot the occlusion map in Python.

In addition, we have generated the cortex and subcortex surface visualizations to show the model's internal emphasis at each ROI (Fig. 4). The regional scores were computed by averaging the normalized occlusion map scores across all AD participants at the same maximum value of ten, and then taking the mean of each ROI of the Desikan–Killiany atlas using a voxel-to-ROI mapping. The visualization and mapping algorithm was achieved through the ENIGMA Toolbox[90].

**Receptive field analysis.** A receptive field based feature map analysis is a good way to decide whether the CNN structure of our UNet is able to cluster the rich voxel-level data from structural MRI into separable channels for the final regression task.[39] Here we present the general ideas for computing the feature map. More technical details are listed in the original paper[39].

Specifically, this is achieved by extracting the neural trace derived from the training data (MRI examinations) of the UNet model, applying the local receptive field analysis method to extract correspondences between quantized features across layers of the model, and visualizing these correspondences as point-clouds and dense volumetric slices with coloring representing different semantic groups (feature phenotype, cognitive score, and so on).

According to the methodology introduced in that paper, the pseudolabels for each layer represent the internal feature clusters derived from the training data[39]. We perform principal component analysis ($d = 3$) and k-means ($k = 3$) on the output distribution of a large sampling ($n = 250$) of training data, and project the pseudolabels in the next layer back to the current layer for visualizing scatterplots. Each patch of input MRI and its corresponding feature map predicted internally by the UNet is quantized into three dimensions using principal component analysis and colored according to an assigned pseudolabel. The pseudolabel here represents a cognitive score grouping ground truth (low, medium, high). By aggregating the quantized patches resulting from several MRI examinations, we discovered a distinctive clustering that separates the cognitive scores of different patients.

The key insight of this work is to use the compositional, convolutional structure of the UNet to better understand and visualize the neural network 'trace' of an input query image.[39] By developing the receptive field analysis (Fig. 4d), we demonstrated that the intermediate representations of UNet correlate strongly with the output prediction, and that these features can be used to assess model performance without ground truth by instead relying on approximation theoretic concepts of data locality, smoothness and organization. This intermediate internal representation of the UNet provided unique insight into how the model itself is organizing the patient's MRI data. This will help model users as well as clinicians understand how the model interprets input MRI data and makes predictive decisions, enabling more informed assessments of the model's performance when run in practice.

## External set-aside testing

To further test the model's effectiveness and robustness, we have conducted out-of-sample testing on an independent testing dataset using data from the DLBS dataset, but perform inference using the exact same ensemble model we trained on the ADNI and HCP–YA cohort previously. In this experiment we chose our best performing ensemble UNet model M9, and applied it to the DLBS data without any further retraining. As this cohort does not include mature patients with AD, it would additionally be quite instructive to assess the applicability of our models trained on the participants on the AD spectrum to this independent healthy aging cohort. Although there are other studies with larger samples, they typically lack a comprehensive ADAS–Cog battery, which constituted our main outcome measure.

**Comparison with alternate models trained on tissue and regional volumetrics.** Our proposed model takes MRI directly as input, yet several previous approaches have used volumetric neuroimaging software to first obtain regional or tissue volumes, followed by ML-based prediction. To demonstrate whether our chosen approach provides a meaningful improvement over these methods, in both accuracy and processing time, we used the XGB ML model that was the most successful amongst all non-DNN model we had explored previously. Inputs to this model were obtained using the widely used FreeSurfer pipeline applied to sMRI data of ADNI participants.

We performed two experiments to achieve benchmark results using volumetric data. First, we trained an XGB model on total volumes reported by Freesurfer of seven broad regions of known importance in AD (ventricles, hippocampus, wholebrain, entorhinal cortex, fusiform gyrus, middle temporal gyrus and total intracranial volume), along with the demographics inputs utilizing the custom gamma loss function. The second model was another XGB model, this time also trained on the cortical and subcortical volumes of all 86 Desikan–Killiany parcellated regions. Thus the first experiment uses pertinent but fewer volumetric features than the second.

## Statistics and reproducibility

Sample sizes were determined by the availability of eligible participants in each cohort after applying predefined inclusion criteria, data linkage requirements and quality control procedures. No statistical methods were used to predetermine sample sizes, however, we used all the available samples in the public databases that met our requirements. Data collection was not randomized, yet there is an inherent randomization step in our chosen CV strategy. No blinding was applied during data collection or analysis. The assumptions of the statistical methods used were assessed where applicable. In particular, gamma distribution assumptions for the customized loss functions were evaluated using standard statistical approaches. Certain test statistics contained in this study implicitly assume normal distribution but this was not formally tested. For ML and DL analyses, robustness and reproducibility were assessed using prespecified criteria, internal CV and external validation across independent datasets. No datapoints were excluded from the analyses beyond predefined cohort eligibility criteria, data quality control procedures and handling of missing data as described in Methods. Consistency of key findings across datasets was used as the primary criterion for assessing reproducibility.

## Reporting summary

Further information on research design is available in the Nature Portfolio Reporting Summary linked to this article.

## Data availability

The public datasets in this study are available from ADNI (https://adni.loni.usc.edu/), HCP–YA (https://www.humanconnectome.org/study/hcp-young-adult) and DLBS (https://openneuro.org/datasets/ds004856/versions/1.2.0). All other data are available from the corresponding author upon reasonable request.

## Code availability

Code for the analyses are available via GitHub at https://github.com/darenma/MultitaskCognition. All analyses were implemented using Python v.3.10. Python dependencies are included in the repository.

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

## Acknowledgements

This study received funding from National Institutes of Health grants: R01AG072753 (A. Raj), R21AG087921 (A. Raj), RF1AG087302 (A. Raj). The funders had no role in study design, data collection and analysis, decision to publish or preparation of the manuscript.

## Author contributions

A. Raj and D.M. conceptualized the study. D.M., A. Raj and Y.Y. designed the study. D.M., C.P. and A. Rajagopal wrote code and executed the study. D.M., A. Raj, Y.Y., A. Rajagopal and Y.I. wrote and edited the paper. D.M., C.P. and A.A. performed data collection and preprocessing. A. Raj supervised the study.

## Competing interests

The authors declare no competing interests.

## Additional information

**Correspondence and requests for materials** should be addressed to Daren Ma, Yang Yang or Ashish Raj.

# Reporting Summary

## Statistics

For all statistical analyses, confirm that the following items are present in the figure legend, table legend, main text, or Methods section.

| n/a | Confirmed | |
|---|---|---|
| ☐ | ☒ | The exact sample size (*n*) for each experimental group/condition, given as a discrete number and unit of measurement |
| ☐ | ☒ | A statement on whether measurements were taken from distinct samples or whether the same sample was measured repeatedly |
| ☐ | ☒ | The statistical test(s) used AND whether they are one- or two-sided *Only common tests should be described solely by name; describe more complex techniques in the Methods section.* |
| ☐ | ☒ | A description of all covariates tested |
| ☐ | ☒ | A description of any assumptions or corrections, such as tests of normality and adjustment for multiple comparisons |
| ☐ | ☒ | A full description of the statistical parameters including central tendency (e.g. means) or other basic estimates (e.g. regression coefficient) AND variation (e.g. standard deviation) or associated estimates of uncertainty (e.g. confidence intervals) |
| ☐ | ☒ | For null hypothesis testing, the test statistic (e.g. *F*, *t*, *r*) with confidence intervals, effect sizes, degrees of freedom and *P* value noted *Give P values as exact values whenever suitable.* |
| ☒ | ☐ | For Bayesian analysis, information on the choice of priors and Markov chain Monte Carlo settings |
| ☒ | ☐ | For hierarchical and complex designs, identification of the appropriate level for tests and full reporting of outcomes |
| ☐ | ☒ | Estimates of effect sizes (e.g. Cohen's *d*, Pearson's *r*), indicating how they were calculated |

*Our web collection on statistics for biologists contains articles on many of the points above.*

## Software and code

Policy information about availability of computer code

| Data collection | No specific code was used to collect the data. The data were downloaded from public datasets. For the dataset availability, check next section. |
|---|---|
| Data analysis | Custom code (python version 3.8) for the UNet and Multi-task MedicalNet algorithms will be available online at https://github.com/darenma/MultitaskCognition. |

For manuscripts utilizing custom algorithms or software that are central to the research but not yet described in published literature, software must be made available to editors and reviewers. We strongly encourage code deposition in a community repository (e.g. GitHub). See the Nature Portfolio guidelines for submitting code & software for further information.

## Data

Policy information about availability of data

All manuscripts must include a data availability statement. This statement should provide the following information, where applicable:
- Accession codes, unique identifiers, or web links for publicly available datasets
- A description of any restrictions on data availability
- For clinical datasets or third party data, please ensure that the statement adheres to our policy

The structural MRI data used in this work are available to researchers via the Human Connectome Project - Young Adults (HCP-YA) data access procedure described at https://www.humanconnectome.org/study/hcp-young-adult/document/extensively-processed-fmri-data-documentation and via the Alzheimer's Disease

Neuroimaging Initiative (ADNI) data access procedure described at http://adni.loni.usc.edu/data-samples/accessdata/.
The ADNI preprocessed MRI data were obtained under account ID daren.ma@ucsf.edu. Source data for the figures will be provided as a Source Data file.

# Research involving human participants, their data, or biological material

Policy information about studies with [human participants or human data](). See also policy information about [sex, gender (identity/presentation), and sexual orientation]() and [race, ethnicity and racism]().

| Reporting on sex and gender | Sex was considered in the study design. We used the "PTGender" column in the ADNI dataset to train the model such that the deep learning structure could capture more biologically relevant underlying features for AD progression.<br>Gender was not considered in this study design. |
|---|---|
| Reporting on race, ethnicity, or other socially relevant groupings | We didn't use race, ethnicity, or other socially relevant groupings in this study. |
| Population characteristics | 1020 HCP-YA Subjects: 488 Females, 532 Males ; Age: range = 22-37 years, mean = 28.7 years<br>1950 ADNI Subjects: 928 Females, 922 Males ; Age: range = 50-92 years, mean = 72.9 years; ADAS-Cog-11: range = 0.7-42.7, mean=10.3.<br>331 DLBS Subjects: 213 Females, 118 Males; Age: range = 25 - 93 years, mean=62.5, ADAS-Cog-11: range = 0-19.3, mean = 5.2 |
| Recruitment | The human data were obtained from three public datasets: ADNI, HCP-YA, and DLBS. |
| Ethics oversight | The ADNI study procedures were approved by the institutional review boards of all participating centers as detailed in this document - https://adni.loni.usc.edu/wp-content/uploads/how_to_apply/ADNI_Acknowledgement_List.pdf. Written, informed consent was obtained from all subjects participating in the study according to the Declaration of Helsinki, and the study was approved by the institutional review board at each participating site. The scientific study protocol of the HCP-YA dataset is approved by the Washington University - University of Minnesota Consortium of the Human Connectome Project (WU-Minn HCP),. Written, informed consent was obtained from all subjects participating in the HCP study. The DLBS study procedures were approved by the institutional review boards of all participating centers as detailed in this document - https://openneuro.org/datasets/ds004856/versions/1.2.0. The study was approved by the institutional review board at each participating site. |

Note that full information on the approval of the study protocol must also be provided in the manuscript.

# Field-specific reporting

Please select the one below that is the best fit for your research. If you are not sure, read the appropriate sections before making your selection.

☒ Life sciences ☐ Behavioural & social sciences ☐ Ecological, evolutionary & environmental sciences

For a reference copy of the document with all sections, see [nature.com/documents/nr-reporting-summary-flat.pdf]()

# Life sciences study design

All studies must disclose on these points even when the disclosure is negative.

| Sample size | All available HCP-YA and ADNI data that passed quality control procedures as described in data exclusions were retained for this study.<br>The segmentation tasks in this work used sMRI images (n = 1020 + 1950) from unaffected subjects (i.e. those who had no diagnosed or self-reported mental illnesses) from the HCP-YA, and neurodegenerative subjects from the ADNI datasets combined.<br>The baseline cognition regression, and baseline diagnosis classification tasks in this work used sMRI images (n = 1950) from the ADNI repository which satisfied our Alzheimer's disease (AD) progression study criterion.<br>The longitudinal cognition regression task involved 1298 subjects with at least two available visits in the ADNI 1, 2, GO, and 3 dataset, meeting all the study criterion. |
|---|---|
| Data exclusions | For the HCP-YA samples, we downloaded in total 1200 from the website. After processing using the FSL FAST toolset, there were 1020 scans with successful segmentation output, and we used these samples for the training of the segmentation tasks, along with the valid ADNI MRIs.<br>For this study we found 2288 subjects in the ADNI 1, 2, 3, and GO metadata dataframe, of which 1950 subjects with (a) current diagnosis status of AD, MCI or Control; and (b) a valid MRI image record for at least one visit.<br>For the longitudinal analysis, we found over 4000 MRI scans for a total of 1952 subjects, who have at least 2 visits recorded, and kept 1298 of them. Cognitive batteries are notoriously error-prone and subject to tremendous inter-rater variability [doi:10.1212/01.wnl.0000434309.85312.19]. Therefore, for the longitudinal cognitive score prediction portion, we found it useful to filter out those subjects whose longitudinal ADAS-Cog data can reasonably be deemed error-prone, resulting in a negative progression in terms of cognition scores.<br>For the set-aside testing dataset from DLBS, we recruited all the data that have a baseline ADAS Cog score, in total 331 subjects from DLBS wave2 and wave3. |
| Replication | A rigorous stratified 9-fold cross-validation procedure was used for nearly all undertaken tasks in this study to ensure reliability of the evaluated performance metrics. The exact same train/validation/test partitions were used across all methods and the performance of all methods was assessed on held-out (unseen) data. All experiments could be successfully validated. |

| Randomization | Randomization was not performed and is not applicable to our study. We did not collect the MRI data but analyzed public data, and we do not study treatment effects. |
|---|---|
| Blinding | Blinding was not performed and is not applicable to our study for the exact same reasons. We did not collect the MRI data, but analyzed public data, and we do not study treatment effects. |

# Reporting for specific materials, systems and methods

We require information from authors about some types of materials, experimental systems and methods used in many studies. Here, indicate whether each material, system or method listed is relevant to your study. If you are not sure if a list item applies to your research, read the appropriate section before selecting a response.

## Materials & experimental systems

| n/a | Involved in the study |
|---|---|
| ☒ | ☐ Antibodies |
| ☒ | ☐ Eukaryotic cell lines |
| ☒ | ☐ Palaeontology and archaeology |
| ☒ | ☐ Animals and other organisms |
| ☒ | ☐ Clinical data |
| ☒ | ☐ Dual use research of concern |
| ☒ | ☐ Plants |

## Methods

| n/a | Involved in the study |
|---|---|
| ☒ | ☐ ChIP-seq |
| ☒ | ☐ Flow cytometry |
| ☐ | ☒ MRI-based neuroimaging |

## Plants

| Seed stocks | *Report on the source of all seed stocks or other plant material used. If applicable, state the seed stock centre and catalogue number. If plant specimens were collected from the field, describe the collection location, date and sampling procedures.* |
|---|---|
| Novel plant genotypes | *Describe the methods by which all novel plant genotypes were produced. This includes those generated by transgenic approaches, gene editing, chemical/radiation-based mutagenesis and hybridization. For transgenic lines, describe the transformation method, the number of independent lines analyzed and the generation upon which experiments were performed. For gene-edited lines, describe the editor used, the endogenous sequence targeted for editing, the targeting guide RNA sequence (if applicable) and how the editor was applied.* |
| Authentication | *Describe any authentication procedures for each seed stock used or novel genotype generated. Describe any experiments used to assess the effect of a mutation and, where applicable, how potential secondary effects (e.g. second site T-DNA insertions, mosiacism, off-target gene editing) were examined.* |

## Magnetic resonance imaging

### Experimental design

| Design type | We used structural MRI scans without any tasks. |
|---|---|
| Design specifications | MRI data preprocessing details are included in the "Online Methods" Sections |
| Behavioral performance measures | N/A |

### Acquisition

| Imaging type(s) | T1-weighetd structural |
|---|---|
| Field strength | 3 Tesla |
| Sequence & imaging parameters | In this study, we included baseline scans of healthy participants from the ADNI-1, ADNI-2, ADNI-Go, and ADNI-3 cohorts. All scans were acquired using an MPRAGE sequence with parameters of TR = 2300/2400–3000 ms, TE ≈ 3 ms, TI = 900–1000 ms, and flip angle = 8°–9°. The images were obtained across multiple sites using scanners from three different manufacturers (GE, Siemens, and Philips). The spatial resolution of ADNI-1 images was 0.9375 mm × 0.9375 mm × 1.2 mm, while that of ADNI-2/Go images was 1 mm × 1 mm × 1.2 mm (Gunter et al., 2009). For the ADNI-3 cohort, the subjects were acquired at 3 T across multiple vendors (GE, Siemens, Philips) using harmonized protocols, with T1-weighted MPRAGE volumes obtained at a resolution of 1 × 1 × 1 mm³ (Gunter et al., 2017). We did not attempt harmonization between phases for ADNI. |
| Area of acquisition | We applied the whole brain cortical scans for the imaging models discussed in the paper. |
| Diffusion MRI | ☐ Used   ☒ Not used |

## Preprocessing

Preprocessing software

FSL-FAST and Python Code

Normalization

Non-linear

Normalization template

MNI

Noise and artifact removal

Certain subjects without an accessible ADNI-Cog-11 score or having missing demographic values in the ADNI dataset was removed.

Volume censoring

N/A

## Statistical modeling & inference

Model type and settings

Multi-task and multivariate Machine Learning and Deep Learning models were applied to analyze the sMRI data with numerical demographic inputs.

Effect(s) tested

R-squared was computed across the study to test if the model results were reflecting the variance from the data. Diagnosis classification accuracy was evaluated for the classification tasks; Dice's Score was evaluated for the segmentation tasks ;mean squared error, R-squared, and Pearson's correlation (between the observed and true values) metrics were evaluated for the regression tasks.

Specify type of analysis:   ☒ Whole brain   ☐ ROI-based   ☐ Both

Statistic type for inference

Whole brain

(See Eklund et al. 2016)

Correction

A permutation feature importance was delivered on the demographic inputs for the two best performing models. No other correction was introduced.

## Models & analysis

| n/a | Involved in the study |
|-----|------------------------|
| ☒ ☐ | Functional and/or effective connectivity |
| ☒ ☐ | Graph analysis |
| ☐ ☒ | Multivariate modeling or predictive analysis |

Multivariate modeling and predictive analysis

This study used 2 dimension reduction methods (UNet and MedicalNet), 4 standard machine learning and 6 customized ensemble deep learning models. Diagnosis classification accuracy was evaluated for the classification tasks; Dice's Score was evaluated for the segmentation tasks ;mean squared error, R-squared, and Pearson's correlation (between the observed and true values) metrics were evaluated for the regression tasks.

