## [Peer Review File · Nature Aging]

Predicting categorical and continuous outcomes of subjects on the Alzheimer's Disease spectrum using a single MRI without PET, cognitive or fluid biomarkers

Corresponding Author: Mr Daren Ma

Version 0:

Reviewer comments:

Reviewer #1

(Remarks to the Author)

1. The paper proposes a novel framework to boost predictive accuracy of machine learning models that predict cognitive scores from brain-MRI scans in Alzheimer's patients. They use a neat idea whereby their model is trained to perform brain tissue segmentation, diagnosis label prediction and ADAS-cog score prediction. The key finding is that by using this multi-task approach and then somehow ensembling the predictions via a meta-model they predict a high proportion of variance in ADAS-cog score. The amount of variance predicted using this framework is markedly higher than the component parts explain separately or more basic benchmarks.

2. The multitask approach is novel to my knowledge and the results seem quite significant, since predicting cognitive performance and indeed change in cognitive performance over time would be useful for Alzheimer's research/treatment in several different ways.

3. The data used were appropriate and standard (ADNI, HCP-YA), though this is a bit limited since the HCP-YA data doesn't have diagnosis or ADAS-cog data, and ADNI has been very widely used. It would have strengthened the paper to use more diverse, real-world datasets. However, as preliminary results, the approach looks very promising. The method is neat and seems valid, though I have issues with the quality of presentation, as I struggled to understand some elements of the network design (more details in #6).

4. The statistics used were appropriate, and although they do not quantify uncertainty in their work, that's quite typical of this kind of machine learning research.

5. The authors conclusions generally align well with the results, so I would say they are valid. There's no particular data on reliability or robustness, adding something towards these would strengthen the paper, but given the innovative and preliminary nature of the work, it's fine as is.

6. Suggested improvements:

- Methods: Did the authors consider a benchmark model where tissue volume (obtained directly from FSL FAST) and demographic data were combined to predict cognition and diagnosis?

- Methods: Data leakage can be a concern with ensemble models. The authors briefly mention using cross-validation at a ratio of 8:1, but I think it's important they provide greater detail. For example, if the training labels from the image segmentation + cognition task were mistakenly carried over to the ensemble cognition task this could erroneously boost performance.

- Methods: The ensemble models isn't clearly explained. It states how the cognition task was integrated into the ensemble, but the ensemble of the segmentation task and diagnosis task was not described.

- Relatedly, Figure 1 is a bit confusing. From the figure, the imaging modules don't seem to do the Diagnosis task. Is this correct? If so, can the authors clarify where the 'Diagnosis' results for models 5-8 in Table 1 come from. The Methods section on the multi-task total loss seems to say that the model is indeed doing 3 tasks, but that doesn't tally with Figure 1.

- Also Figure 1, is the imaging module part of the ensemble? If the above is correct (i.e., Imaging modules don't do diagnosis), this implies that the ensemble is cognition/cognitive-change prediction only. This is fine, but not very clear in the figure. Or perhaps the latent variables from the segmentation are trained to predict the all 3 multitask outcomes and then ensembled with the demographic and volume values? Can the authors clarify this and point out if the latent-variable regression results appear in Table 1.

- If my understanding of Figure 1 is correct, it feels like the segmentation task is being used as a feature extractor pre-task, rather than it all being a simultaneous multi-task framework. I'm very open to being corrected on this, but the precise description of the approach throughout the manuscript would benefit from more clarity, particularly in Figure 1 and the Methods.

- Results>Segmentation. FSL FAST doesn't produce ground truth brain tissue segmentations. I suggest the authors use 'silver standard' instead (as they do on page 14 in fact).

- Figure 2F. The labels in the figure are a bit confusing. Should they be 10:MN+XGB and 9:UNet+XGB? Worth trying to be consistent within the same figure.
 - Primary target 2: Cognitive scores (Longitudinal) Section B The authors mention using the z-scores of the segmented tissue volumes compared to healthy controls. This requires more detail, such the sample size, age and sex of the controls, and information on the inclusion/exclusion criteria to be defined as 'healthy'. And was this a simple z-score or was a normative model of some description used?
 - The demographic details in supplementary table 1 are very relevant to the study and I recommend them being in the main manuscript.
7. References are appropriate
8. The prose is generally fine throughout, but the precise description of the Methods need quite a lot more clarity.

(Remarks on code availability)

Reviewer #2

(Remarks to the Author)

In this study, Ma et al. trained and tested deep learning models that can produce tissue segmentation, determine if subjects have AD dementia or not, and predict longitudinal cognitive decline as measured by ADAS-Cog with high accuracy. This is done solely on the basis of baseline structural MRI and demographic data from ADNI, with no other biomarkers or variables that have been shown in some cases to boost model performance, thereby supposedly increasing translational utility in clinical setting where assessment resources are limited. I have mixed feelings about this work: On the one hand, it's really amazing to see the great potential of AI/DL as a tool to revolutionize medical decision-making, which I have no doubt will prove invaluable for clinicians, patients, and their families in the long run. On the other hand, this manuscript reads too much like a sales pitch, undermines some of the current standards in the field, and conveys unclear and mixed messages regarding the true clinical and practical utility of MRI-based diagnosis and prognostication via DL. Below, I provide my comments on these aspects in detail, which I hope the authors will find useful in improving the message conveyed by this manuscript.

1. Although I find the tone of some of the authors' statements to be unnecessarily adversarial (see my comment #2), I generally find the Introduction section to be easy to follow and well written. However, the most serious concern I have with this work is that it focuses too much on the literature on machine/deep learning, while largely ignoring a growing body of evidence in the literature on clinical prognostication in AD. Just within the past several years, numerous studies have used examined the utility of imaging and fluid biomarkers, indices of cognitive/functional impairment, and other characteristics for predicting subsequent cognitive decline across a broad symptomatic and syndromic spectrum of AD (see a few references below, and those therein). These studies have demonstrated which features have relatively more clinical prognostic value (e.g., imaging > cognition; tau PET > MRI or fluid), and in the case of imaging features, which brain region(s) are more important than others in terms of their contribution, which have profound implications for therapeutic interventions (e.g., by stimulating such regions via neuromodulation). I believe it is essential for the authors to contextualize their work in this literature, to more clearly highlight the novel contribution of this study, not just in terms of the sophistication and accuracy of their DL models, but also its clinical relevance beyond what has already been established using perhaps simpler yet certainly powerful and robust tools.

Groot C, Smith R, Collij LE, et al. Tau Positron Emission Tomography for Predicting Dementia in Individuals With Mild Cognitive Impairment. *JAMA Neurol.* 2024;81(8):845-856. doi:10.1001/jamaneurol.2024.1612

Katsumi Y, Howe IA, Eckbo R, et al. Default mode network tau predicts future clinical decline in atypical early Alzheimer's disease. *Brain.* Published online October 16, 2024. doi:10.1093/brain/awae327

Smith R, Cullen NC, Pichet Binette A, et al. Tau-PET is superior to phospho-tau when predicting cognitive decline in symptomatic AD patients. *Alzheimers Dement.* 2023;19(6):2497-2507. doi:10.1002/alz.12875

2. The authors' language in some parts of the manuscript, as well as its overall organization, feels quite dehumanizing, putting too much emphasis on the technical advantages of AI/DL while undermining its translational utility in the context of the tools and standards established in the literature and clinical care. The authors note in the Abstract that current tools for cognitive assessment are "subjective, time-consuming, and operator-sensitive", while making a similar statement in the supplement that cognitive assessments are "error-prone" and "subject to tremendous inter-rater variability". Yet, ADAS-Cog and longitudinal changes in this measure were the outcome variables of this study. What's the point of using fancy algorithms to predict something that the authors believe to be psychometrically unreliable and its clinical validity questionable?

3. The authors note that available software packages used to process MRI data, like FreeSurfer, FSL, and ANTs, are time-consuming and computationally demanding. These resources are open-source, have been extensively documented, continuously updated, and have active community of users worldwide, making it very easy for beginners to get started with image processing with minimal top-down knowledge. If the authors feel strongly about the importance of speed and efficiency in real-world clinical prognostication and decision-making, it should be discussed explicitly just how much time their DL-based tools would save. In clinical setting, though, when would we ever need to formulate clinical prediction within seconds after the acquisition of MRI?

4. It should be made clear as early as possible in the Results section (or toward the end of Introduction) which subject population was examined. The way this section is written now, the reader wouldn't see really basic characteristics of the sample (e.g., age, cognitive functioning) until later in the manuscript. "ADNI" and "HCP" mean different things to different people, so I'd suggest the authors provide basic cohort characteristics in the opening paragraph of the Results section so the reader can better understand the nature of data that were analyzed in this work.

5. There is variability in imaging sequences and parameters across different phases of ADNI. This should be explicitly acknowledged, along with some details about how the authors accounted for this variability or not in their analytical approach.

6. I think it would be fantastic if, as they say, the authors' algorithm can produce tissue segmentations "within seconds" that have comparable quality to those estimated more traditionally using FSL FAST. This should be backed up by empirical evidence and reported in the Results section, instead of just mentioning it in passing in the Introduction.

7. Table 1: Please provide the breakdown by AD dementia, MCI, and CN per set.

8. The distinction between "AD" and "non-AD" mentioned on page 5 is ambiguous and needs to be more thoroughly explained in the Results section. It seems that the authors rather mean "AD dementia" vs. "non-AD dementia (i.e., CN or MCI)", but then saying that MCI represents a "non-clinical group" (p. 14) is simply incorrect.

9. Also related to the diagnostic classification task, it's unclear what it means for the model to use both ADNI and HCP data as inputs, when the latter contained only young adults and no information about dx. Could the authors clarify? Ditto the analysis on predicting baseline cognition. This sounds contradictory to what the authors note in Online Methods, that the HCP YA data were used only for the purpose of augmenting the segmentation task.

10. Feature importance: Results shown on Fig. 4B deserve more investigation. I find it quite surprising and intriguing that cortical gray matter has little contribution to predicting cognition. If I'm reading this brain map correctly, it appears that the thalamus and some of the brainstem nuclei have the most significant contribution in predicting baseline cognition. This makes little sense in the context of clinical prognostication in AD just in terms of its pathological-degenerative cascade established in the field, provides very limited implications for possible interventions/clinical trials, and again makes me really wonder what the additional novel contribution this work has beyond what's been already shown in the literature.

11. Data augmentation using image preprocessing: This section requires further clarification to be more accessible by the reader. Am I correct in that the authors manipulated some of the existing MRI data to generate synthetic images and treated them as if they had been acquired from new, completely independent subjects? How many such cases were included, and would this pose a problem in terms of input dependency? Please also comment on what processing steps, if any, were implemented to input MRI data in general. Some MRI images have different resolution than 1 mm isotropic voxels – was resampling the only step that was involved? Any intensity normalization?

(Remarks on code availability)

No code is provided in this submission.

Reviewer #3

(Remarks to the Author)

I enjoyed reviewing the manuscript and think it is interesting and novel. Predicting cognition from MRI is a relevant task. While the multimodality aspects of the paper are very intriguing, they are to some extent expected and prevent to fully judge the impact of the technological advancements the authors propose. The diagnosis and segmentation use cases seem of lower novelty and impact to me. I think the paper could use some work to foster a better understanding of details and the experimental set ups.

Major points:

Figure 1 is not clearly understandable. The authors talk about a 'regression layer' and the tree-based 'regression module' are those the same? Or is the regression layer just a linear layer to which the low-dimensional representations are fed? Which multitask predictions does the regression layer solve (the three shown on the right? What's the dimensionality then?)? Why are the latent features fed into the ensemble model and more concrete features (voxel counts etc) to the boosting algorithm?

I don't fully understand the value in having the model perform the three described tasks (segmentation, diagnosis, predicting cognition) in one approach. I understand that all of these can be of interest in different scenarios but seldomly they are needed all together. Is a model solely dedicated to predicting cognition worse? The authors refer to previous works but have not provided a head-to-head comparison for their datasets.

"While the segmentation task serves [...] to enable efficient learning in a data-scarce setting" there is no data shown in the manuscript to merit that conclusion. I see where the authors are coming from but without evidence, this remains speculative.

I would love to see a comparison not only against an XGB with demographics only but an XGB with FreeSurfer generated features (volumes, thickness). I think this would be a fairer comparison, since now the different access to information is biasing the model comparison and thus it's unclear whether the gain in performance is due to additional features/types of

information or technological advancements made by the authors (plus, the data is readily available for ADNI and easy to access).

I would like to see further regression metrics besides the R2, to judge the clinical value of the predictions. The MAE and MAPE would be interesting to look at.

Fig 3A can only be interpreted when knowing how accurate the determined empirical alphas actually represent the trends in the data. I prefer the results shown in Fig 3 B and C.

It would be interesting to have a Supplementary figure that gives an overview of the predicted alpha-parameterized trajectories and the empirical ones.

For Fig 4 A, y-axes should cover the same range. There is also a mismatch in dimensionalities as the image nets will feed significantly more features into the XGBs.

It would have also been interesting to see how much the latent representations add to the XGB over the explicit features calculated by the image nets.

I would have liked to see a proper external validation on independent data (for example training on ADNI, testing on another cohort).

“Both scale parameters σ and γ are estimated directly from empirical data at the pre-processing stage, without further learning.”

- The parameters need to be calculated inside the CV. Otherwise you will run into overfitting through data leakage
- How would this transfer to other cognitive outcomes that may not follow the same distribution (i.e., CDR and CDR Sum of Boxes scores, the latter of which is often used in clinical trials for AD)?

It is not clear how class imbalances were attributed for when presenting the accuracy of the diagnosis prediction.

“Accuracy increased significantly to 94.56% when we combined the UNet and XGB models into a multi-task approach, leveraging both ADNI + HCP inputs and demographics”, does this imply that the results were only produced on one of these data sources? If so, the performance gain is not necessarily due to the technical contributions of the authors.

“Data augmentation using image pre-processing”

The authors should provide more specific information in the Supplementary Material.

From the methods it is not always clear based on which performance decisions about model architecture were made (e.g., deciding for the XGB module and going for the Resnet50 instead of 10). Such decisions should be made solely based on the CV results.

Are the XGB and DNN trained in v? The L_DNN and L_XGB makes it seem they are not. I would be interested to see whether an end-to-end training makes the system better (as image net-derived features are specifically catered to the cognitive prediction task). Or is L_Cog in L_DNN derived from the XGB output (I guess its from the ‘regression layer’ mentioned in Fig 1)? The entire set-up could be described a bit more clearly.

Minor:

- Regarding accurate language, it seems like the authors here are not targeting AD as it is defined nowadays but the clinical syndrome of AD, being AD dementia.
- I think it would be good to mention that the “3D ResNet50 architecture” mentioned in the introduction is later referred to as MedicalNet
- In the legend of Table 1 it is written that a 10-fold CV was conducted. In the Segmentation section it says 5-fold
- Talking about “excellent” results reads a bit like confirmation bias to me. I would prefer no judgement about the results in the Results section and letting the numbers speak for themselves.
- Typo: “r4ecently”
- “Tencent” spelling is inconsistent

(Remarks on code availability)

Version 2:

Reviewer comments:

Reviewer #1

(Remarks to the Author)

- Overall, the authors have done a very good job on responding to the reviewers’ comments, and the manuscript is definitely improved, particularly in terms of clarity and language.
- The description of the ensemble approach is improved, though I think needs a bit more detail. Firstly, I haven’t come across an ‘R^2 ensemble’ approach before. I assume this serves to weight the base models by the variance explained by the

Imaging module and Non-imaging module. So if the modules had similar R^2 , then actually this ensemble would have limited effect? I think it would be useful to clarify exactly what the R^2 values from the cross-validation were for each of the modules, so that the effect of the ensemble is more transparent. This is an important point because one possibility for the improvements after the ensemble is simply that the network was trained for more epochs.

- Building on this, generally there is a lack of mention of the numbers of epochs used for training, and whether over-fitting was evaluated. Clearly some overfitting occurred, as the performance in the additional DLBS data explains considerably less variance than on ADNI.

- In fact, considering the DLBS dataset, it would have been good to test all their models (i.e., M1-M10) on the DLBS data, not just M9. This would help quantify whether the top-line ensemble performance in the ADNI is due to ensemble over-fitting or whether the performance improvement is maintained out-of-sample.

(Remarks on code availability)

Reviewer #2

(Remarks to the Author)

The authors have thoroughly considered and addressed most of the concerns I noted in my initial review of this work.

Regarding Comment #5, the authors' statement regarding parameter variability across ADNI phases is incorrect. It wasn't until ADNI3 when MPRAGE was acquired using 1-mm isotropic voxels. ADNI Go/ADNI2 had slice thickness of 1.2 mm.

Perhaps the authors are confusing "phases" with "sites". In any case, for full transparency, I still suggest the authors report which ADNI phases the current subject sample represented. Even if the authors chose not to do so, their current statement about all ADNI phases associated with 1-mm iso MPRAGE is inaccurate and I'd advise them to drop it.

(Remarks on code availability)

Version 3:

Reviewer comments:

Reviewer #1

(Remarks to the Author)

- Good to see that they've corrected the confusion matrix, though this has raised another important question. Table 1 and Table 2 say the ADNI test set $n=195$ and the results in Table 2 are from the 'set aside testing', yet the results in Table 2 are exactly the same as the confusion matrix, where the $n=1950$ (i.e., M9 accuracy in Table 2 is 94.56%, which is the same as the updated confusion matrix in Figure 2 and Supplementary Table 7). It's not impossible to get identical results to the 2 decimal place when evaluating on $n=195$ vs. $n=1950$, but seems unlikely. Can the authors confirm what the sample size used in Table 2 was?

- Either way, it's unclear how the predictions were generated for the $n=1950$ confusion matrix in Figure 2/Supplementary Table 7. On page 12, they describe the validation scheme as either 9:1 or 8:1:1. Both imply that the results are only being evaluated on the 'set-aside' test set (i.e., $n=195$), not on all the ADNI data ($n=1950$). This section needs updating to explain how they generated predictions that populate the updated confusion matrix and supplementary table 7. I realise they used cross-validation within the 90% training data, but that shouldn't really be reported as 8:1:1. More importantly, it also means that 90% of the predictions were generated within the training cross-validation while the remaining 10% were truly set side. However, the results in Figure 2/Supplementary Table 7 are reported all together, ignoring this distinction between how the training and test were split, which is not really appropriate. Moreover, it does not solve the issue of the identical results in the confusion matrix and Table 2, which purportedly come from different (overlapping) test sets.

- I think it is essential that this issue is clarified before publication. I hope this is not the case, but it's possible that there have been some errors in how the results were generated which could have caused data leakage and inflation of performance.

(Remarks on code availability)

Version 4:

Reviewer comments:

Reviewer #1

(Remarks to the Author)

Good to see that the authors have updated the manuscript a bit, but I still don't think it's appropriate to merge results from CV and held-out testing. CV is known to inflate performance relative to hold-out, so they really should be reported separately. Also, Figure 2 remains a bit confusing with some of the results presented just on the held-out (e.g., panel G) with others on the merged CV and held-out test data (e.g., panel D). While it explains this in the legend, I don't see the motivation for doing it.

(Remarks on code availability)

Dear Editor,

Thank you for arranging a review of our manuscript, and for giving us a chance to address prior critiques. We have gone through reviewer critiques very carefully, and our assessment is that they have broadly been quite positive about our study. They have raised several important points, which were all eminently addressable by leveraging ongoing work in our laboratory. In this revised manuscript, we have addressed their major concerns, as outlined below. We also appreciated your specific guidance on what would need to be implemented in such a revision, and we have been able to entirely implement them.

This Revision contains substantial new passages throughout, intended to address specific reviewer comments. These are highlighted in the revised manuscript.

Here we summarize the major changes for your reference: one new main Figure with new results, three edited main Figures with updated results or layouts, two main edited tables with more inclusive data. These changes roughly followed three key aspects requested by the editors during the last round:

1. **External set-aside testing data.** We collected 331 subjects with sMRI scans and ADAS Cog scores from the unrelated and independent Dallas Lifespan Brain Study dataset and tested the UNet model performance on them, without further retraining. The testing R^2 scores reached 0.62 on unseen data, which we found very compelling. This result, prompted by both editor and reviewers, fully serves to demonstrate the utility and generalizability of our tool on wider contexts.
2. **Non-imaging FreeSurfer-based benchmark.** Reviewers had requested that we compare our approach quantitatively with more established methods in the field that have used secondary biomarkers obtained from volumetric software applied to raw MRI. We collected the Freesurfer volumetric data from the ADNI dataset, and built two XGB models to predict the baseline cognition using Freesurfer volumetric data. The results were added to the Results section and in a new Figure, and conclusively establish the superior performance of our proposed approach.
3. Altering the language around AI/ML in order to “humanize” it; adding vital missing context around the role of biomarkers in the dementia field other than pure machine learning models; watering down several overly strong claims.

The following contains a line-by-line response to the reviewers’ advice. We are confident that these revisions have made our manuscript far more compelling than the original, and will hopefully appeal to the reviewers and to your readership. We continue to strongly believe this to be a highly impactful study, with wide potential for clinical translational applicability in aging populations. We sincerely present this revised manuscript to be reconsidered by Nature Aging.

Best,

Daren Ma, Yang Yang and Ashish Raj, on behalf of all co-authors

Detailed Response to Reviewers' Comments

Reviewer #1

1. The paper proposes a novel framework to boost predictive accuracy of machine learning models that predict cognitive scores from brain-MRI scans in Alzheimer's patients. They use a neat idea whereby their model is trained to perform brain tissue segmentation, diagnosis label prediction and ADAS-cog score prediction. The key finding is that by using this multi-task approach and then somehow ensembling the predictions via a meta-model they predict a high proportion of variance in ADAS-cog score. The amount of variance predicted using this framework is markedly higher than the component parts explain separately or more basic benchmarks.

2. The multitask approach is novel to my knowledge and the results seem quite significant, since predicting cognitive performance and indeed change in cognitive performance over time would be useful for Alzheimer's research/treatment in several different ways.

We thank you for these encouraging and positive comments.

3. The data used were appropriate and standard (ADNI, HCP-YA), though this is a bit limited since the HCP-YA data doesn't have diagnosis or ADAS-cog data, and ADNI has been very widely used. It would have strengthened the paper to use more diverse, real-world datasets. However, as preliminary results, the approach looks very promising. The method is neat and seems validity, though I have issues with the quality of presentation, as I struggled to understand some elements of the network design (more details in #6).

We agree with this and other comments requesting additional datasets. As you will note below, we have now introduced additional independent sources of data to fully assess out-of-sample validation.

4. The statistics used were appropriate, and although they do not quantify uncertainty in their work, that's quite typical of this kind of machine learning research.

5. The authors conclusions generally align well with the results, so I would say they are valid. There's no particular data on reliability or robustness, adding something towards these would strengthen the paper, but given the innovative and preliminary nature of the work, it's fine as is.

We have thoughtfully considered your suggestion. We have added the standard deviation of the cross validated R-squared scores in Table 2 (previously Table 1), which hopefully helps to evaluate the robustness of our ML models and therefore emphasize reliability of this study.

6. Suggested improvements:

- Methods: Did the authors consider a benchmark model where tissue volume (obtained directly from FSL FAST) and demographic data were combined to predict cognition and diagnosis?

We thought this advice was invaluable and implemented the tissue volume benchmark modeling in the revised manuscript. (See Figure 5 C, D, and E.) We used the FreeSurfer data instead of FSL FAST ones, as suggested by R3, but the two volumetric methods are essentially equivalent. We found that indeed our proposed approach is superior to these benchmarks (R^2 of 0.54 vs 0.83 from our

model). This is especially useful in a practical setting, since our model inference takes a few seconds while performing Freesurfer volumetrics would require about 8 minutes per subject.

- Methods: Data leakage can be a concern with ensemble models. The authors briefly mention using cross-validation at a ratio of 8:1, but I think it's important they provide greater detail. For example, if the training labels from the image segmentation + cognition task were mistakenly carried over to the ensemble cognition task this could erroneously boost performance.

This has been addressed in Methods - Cross-validation and set-aside data splits. To clarify our efforts of cross validation, we performed a train-test split at a ratio of 9:1. This 10% testing set is set aside and never used in the training or validating processes, until we report the testing performance of each model. On the remaining 90% of samples, we performed 9-fold cross validation for all the models.

To eliminate data-leakage, we made sure that the patient ID information was preserved across all stages of processing but dropped before any model fitting. Plus, we applied the same random seed in performing the train-validation splitting.

- Methods: The ensemble models aren't clearly explained. It states how the cognition task was integrated into the ensemble, but the ensemble of the segmentation task and diagnosis task was not described.

We have added a comprehensive subsection explaining the ensemble approach and the specific formulation of ensemble loss in Methods section. We clarified that the diagnosis task had the same ensemble approach as the cognition task, with the only difference being the loss function. The segmentation task had no ensemble because the imaging module handled it independently. However, we have clarified that the full model is retrained end to end with the ensemble loss, including the UNet layers that were originally trained on the segmentation task.

- Relatedly, Figure 1 is a bit confusing. From the figure, the imaging modules don't seem to do the Diagnosis task. Is this correct? If so, can the authors clarify where the 'Diagnosis' results for models 5-8 in Table 1 come from. The Methods section on the multi-task total loss seems to say that the model is indeed doing 3 tasks, but that doesn't tally with Figure.

We apologize for the confusion. The imaging module is certainly being used for all tasks, but this was not conveyed properly in our schematic. To clarify this, we have remade Figure 1 so that it better reflects the model design. Note that the imaging modules do produce Diagnosis results, and that's where the Table 2 (previous Table 1) results are from. The loss function in the Methods section provides the mathematical details of this approach. We hope the reworked illustrations in Figure 1 cleared this up.

- Also Figure 1, is the imaging module part of the ensemble? If the above is correct (i.e., Imaging modules don't do diagnosis), this implies that **the ensemble is cognition/cognitive-change prediction only**. This is fine, but not very clear in the figure. Or perhaps the latent variables from the segmentation are trained to predict the all 3 multitask outcomes and then ensembled with the demographic and volume values? Can the authors clarify this and point out if the latent-variable regression results appear in Table 1.

This is a similar problem as the last one, and we believe the refined Figure 1 will make these all clear. The ensemble is in fact on these tasks: baseline cognition, baseline diagnosis, plus the longitudinal cognition change rate. We have also changed the caption in that Figure to minimize potential confusion or misunderstanding.

In terms of the latent-variable results, all the UNet and MedicalNet models in Table 1 provide their regression prediction through feeding the latent variables (i.e. the low-dimensional features from the bottom of the UNet or MedicalNet) into a fully connected NN – this is now clearly depicted in Figure 1. Thus, these latent variables from the segmentation task are then used to predict all 3 multitask outcomes and then ensembled with the XGB module which takes in the demographic and volume values.

- If my understanding of Figure 1 is correct, it feels like the segmentation task is being used as a feature extractor pre-task, rather than it all being a simultaneous multi-task framework. I'm very open to being corrected on this, but the precise description of the approach throughout the manuscript would benefit from more clarity, particularly in Figure 1 and the Methods.

With the clarifications and reworked figure above, we this issue stands clarified. The segmentation task serves a bigger role than a “feature extractor”, because the imaging model that was trained to achieve segmentation was then further retrained end-to-end on the other tasks. Hence segmentation is not only a valuable secondary product, but also integral to the success of the other tasks.

- Results>Segmentation. FSL FAST doesn't produce ground truth brain tissue segmentations. I suggest the authors use 'silver standard' instead (as they do on page 14 in fact).

We agree that FAST segmentations should not be addressed as “ground truth” or “gold standard”. We have replaced the wording to “silver standard” throughout the manuscript.

- Figure 2F. The labels in the figure are a bit confusing. Should they be 10:MN+XGB and 9:UNet+XGB? Worth trying to be consistent within the same figure.

We intended to make the subplot 2F consistent with the coloring and words of Figure 3 as well, so it was presented that way. Thanks to your advice, now we have added model numbers as well to increase clarity.

- Primary target 2: Cognitive scores (Longitudinal) Section B The authors mention using the z-scores of the segmented tissue volumes compared to healthy controls. This requires more detail, such the sample size, age and sex of the controls, and information on the inclusion/exclusion criteria to be defined as 'healthy'. And was this a simple z-score or was a normative model of some description used?

According to your suggestion, we have added the details to this subsection. The healthy controls are selected from the same ADNI baseline cohort, whose diagnosis in the original dataset are labeled as HC. We have updated the sample size of 1388 total scans meeting our exclusion criteria. The exclusion criteria are the same as elsewhere in this study. We used the simple z-score in this approach.

- The demographic details in supplementary table 1 are very relevant to the study and I recommend them being in the main manuscript.

We have moved this and the corresponding text to the main manuscript.

7. References are appropriate

8. The prose is generally fine throughout, but the precise description of the Methods need quite a lot more clarity.

After having implemented on your detailed suggestions above, we may confidently note that the Methods presentation is now far clearer and precise.

Reviewer #2

In this study, Ma et al. trained and tested deep learning models that can produce tissue segmentation, determine if subjects have AD dementia or not, and predict longitudinal cognitive decline as measured by ADAS-Cog with high accuracy. This is done solely on the basis of baseline structural MRI and demographic data from ADNI, with no other biomarkers or variables that have been shown in some cases to boost model performance, thereby supposedly increasing translational utility in clinical setting where assessment resources are limited. I have mixed feelings about this work: On the one hand, it's really amazing to see the great potential of AI/DL as a tool to revolutionize medical decision-making, which I have no doubt will prove invaluable for clinicians, patients, and their families in the long run. On the other hand, this manuscript reads too much like a sales pitch, undermines some of the current standards in the field, and conveys unclear and mixed messages regarding the true clinical and practical utility of MRI-based diagnosis and prognostication via DL. Below, I provide my comments on these aspects in detail, which I hope the authors will find useful in improving the message conveyed by this manuscript.

We thank you for these highly encouraging remarks. Your critical comments have been highly constructive and as noted below have been squarely addressed in the revision.

We have taken to heart the need to be more balanced and to remove the "sales pitch". Oftentimes, the textual language can be interpreted by reasonable readers in a manner not intended by the author - we believe this to have been the case. Nonetheless, we take full responsibility for the impressions our language has caused, and have tried hard to address this.

1. Although I find the tone of some of the authors' statements to be unnecessarily adversarial (see my comment #2), I generally find the Introduction section to be easy to follow and well written. However, **the most serious concern** I have with this work is that it focuses too much on the literature on machine/deep learning, while largely ignoring a growing body of evidence in the literature on clinical prognostication in AD. Just within the past several years, numerous studies have used examined the utility of imaging and fluid biomarkers, indices of cognitive/functional impairment, and other characteristics for predicting subsequent cognitive decline across a broad symptomatic and syndromic spectrum of AD (see a few references below, and those therein). These

studies have demonstrated which features have relatively more clinical prognostic value (e.g., imaging > cognition; **tau PET > MRI or fluid**), and in the case of imaging features, which brain region(s) are more important than others in terms of their contribution, which have profound implications for therapeutic interventions (e.g., by stimulating such regions via neuromodulation). I believe it is essential for the authors to contextualize their work in this literature, to more clearly highlight the novel contribution of this study, not just in terms of the sophistication and accuracy of their DL models, but also its clinical relevance beyond what has already been established using perhaps simpler yet certainly powerful and robust tools.

Groot C, Smith R, Collij LE, et al. Tau Positron Emission Tomography for Predicting Dementia in Individuals With Mild Cognitive Impairment. *JAMA Neurol.* 2024;81(8):845-856.
doi:10.1001/jamaneurol.2024.1612

Katsumi Y, Howe IA, Eckbo R, et al. Default mode network tau predicts future clinical decline in atypical early Alzheimer's disease. *Brain.* Published online October 16, 2024.
doi:10.1093/brain/awae327

Smith R, Cullen NC, Pichet Binette A, et al. Tau-PET is superior to phospho-tau when predicting cognitive decline in symptomatic AD patients. *Alzheimers Dement.* 2023;19(6):2497-2507.
doi:10.1002/alz.12875

Absolutely, there are several valuable non-imaging biomarkers that have been reported for their predictive value in AD. In fact, this was one of the key motivations behind our work: that currently MRI by itself is not a very good biomarker *compared to other modalities and biomarkers*. Given our focus on using a single MRI as input to our model due to its clear clinical use case, we did not previously present a thorough balanced view of other biomarkers. Nor is it possible to fairly cover the huge breadth of biomarker research in AD, and their biological implications, in the current manuscript, which has a narrow focus on machine learning on MRI. It was far from our intent to support our chosen model as a substitute for other biomarkers. If anything, our point was more about the deficiencies of current ML methods applied to MRI, instead of current biomarkers, whose clinical and scientific utility is not in doubt. There is tremendous scope for innovation that combines multiple techniques, and perhaps the best way forward is to bring both AI and conventional methods together. Our study was in fact designed to take a step in that direction, by introducing domain knowledge of the AD context to AI methods. We have clearly highlighted this in revised Introduction, quoted below:

In Introduction:

“2. Meta-analytical studies report that MRI has little incremental value for classification tasks in comparison with non-imaging clinical and cognitive biomarkers [9, 20]. It was the least predictive amongst multiple modalities [8, 9], and was not supported for early diagnosis of dementia as a stand-alone add-on test [21]. DNNs give significantly better classifier performance when multiple expensive neuroimaging modalities (MRI and PET), multiple visits, and clinical, cognitive and fluid biomarkers are used together. Very few instances report good classifier accuracy using a single MRI alone, without PET, cognitive or clinical biomarkers [6, 9, 20, 22].”

Further:

“However, it is important to note that our approach is not meant as a substitute for other biomarkers and modalities already reported in the vast AD literature. On the contrary, successful AI-based

integration of MRI may open the door to further enhancements that include additional biomarkers when available.”

These passages cite the papers you noted, in addition to a few others we were able to evaluate. Further, based on your comment we are adding a new section in Discussion on the value of non-MRI, especially non-imaging, biomarkers:

Scientific versus predictive value of AD biomarkers

In recent years, numerous studies have examined the utility of various biomarkers such as hippocampal shape features from MRI [41], fMRI features [42,43], fluid biomarkers, neurocognitive batteries and APOE status [48], and PET [44-47] for predicting subsequent cognitive decline across the broad spectrum of AD. The predictive value of biomarkers in these studies is typically assessed statistically, with a focus on identifying which features have the most clinical prognostic value. Tau-PET in particular is a highly specific and sensitive biomarker of dementia conversion [49-51], although less so for predicting cognitive scores. We discussed above many such biomarkers in the context of their utility in machine learning predictive models, yet their value extends beyond, into understanding which types of biomarkers (e.g. blood or CSF protein levels, specific brain regions like hippocampus) are more important than others in terms of their contribution. Insight into these issues can have profound implications for mechanistic insights and therapeutic interventions (e.g., by stimulating such regions via neuromodulation). Hence the present contribution should be evaluated narrowly in terms of predictive and utilitarian value in a clinical context, rather than the broader landscape of AD biomarkers.

Thus, our goal is to improve the predictive ability of MRI as opposed to PET and non-imaging biomarkers the reviewer has specified. We hope the new language noted above has helped clarify our stand and ameliorated your concern.

2. The authors' language in some parts of the manuscript, as well as its overall organization, feels quite dehumanizing, putting too much emphasis on the technical advantages of AI/DL while undermining its translational utility in the context of the tools and standards established in the literature and clinical care. The authors note in the Abstract that current tools for cognitive assessment are “subjective, time-consuming, and operator-sensitive”, while making a similar statement in the supplement that cognitive assessments are “error-prone” and “subject to tremendous inter-rater variability”. Yet, ADAS-Cog and longitudinal changes in this measure were the outcome variables of this study. What's the point of using fancy algorithms to predict something that the authors believe to be psychometrically unreliable and its clinical validity questionable?

We understand the reviewer's concern regarding the tone of some statements. We have modified the language to be less adversarial and have removed any terms that could be perceived as “dehumanizing”. We sincerely believe that rapid advances in AI/ML in the aging field - of which our study may be an exemplar - do not invalidate previous more conventional analyses, instead they serve to open new possibilities that were perhaps under-served. It is far from our intent to support our chosen model as a substitute for other biomarkers. There is tremendous scope for innovation that combines multiple techniques, and our study was specifically intended to combine domain knowledge and modern AI.

About ADAS-Cog scores: we have removed from our language the unintended implication that ADAS-Cog is unreliable or does not have clinical validity. This would be, as the reviewer points out, exactly contrary to our stated goals. We have deleted the phrases noted by reviewer. We do wish to

highlight to the reviewer that in community clinical settings, a patient would not normally have access to neuropsych testing yet would benefit from a reasonable and speedy way to assess their cognitive status.

3. The authors note that available software packages used to process MRI data, like FreeSurfer, FSL, and ANTs, are time-consuming and computationally demanding. These resources are open-source, have been extensively documented, continuously updated, and have active community of users worldwide, making it very easy for beginners to get started with image processing with minimal top-down knowledge. If the authors feel strongly about the importance of speed and efficiency in real-world clinical prognostication and decision-making, it should be discussed explicitly just how much time their DL-based tools would save. In clinical setting, though, when would we ever need to formulate clinical prediction within seconds after the acquisition of MRI?

We do not have a strong opinion on this matter; whether current volumetric tools are useful is best left to the end user to determine. We simply wish to provide other users the opportunity to benefit from implicit spatial representations in the brain, learned by our proposed model, without requiring expertise of these computational pipelines. Nonetheless, the suggestion to provide quantitative metrics of speed performance is well taken. In the Revision we supply these numbers (see Supplementary Table 4 and text around new Figure 5), which shows that the single sample inference time is around 10 seconds, faster than the 7 minute average analyzing time of FSL FAST. We have also now, based on another reviewer's suggestion, implemented a comparative analysis of predictive power of Freesurfer volumetrics in contrast to our DNN. Here too we were able to show compelling improvement in performance.

The reviewer's points about the continuing relevance of processing pipelines are well taken. We have inserted the following text in and Discussion→Translational Potential:

"These processing tools are well-established, open-source, extensively documented, continuously updated, and have active and dedicated community of users worldwide. Hence, they will continue to be vital tools for many researchers. The current study simply provides other users, especially clinicians, the opportunity to benefit from implicit spatial representations in the brain, learned by our proposed model, without requiring expertise of these computational pipelines. We reported meaningful gains in speed and performance over these pipelines, which could prove valuable in specific contexts, e.g. in developing a quick prediction of cognitive impairment in the clinic prior to referring the patient to a more advanced imaging lab and/or full neuroradiology report. However, clinical utility in practice will depend on specific use cases and environments and will require careful assessment in future studies."

4. It should be made clear as early as possible in the Results section (or toward the end of Introduction) which subject population was examined. The way this section is written now, the reader wouldn't see really basic characteristics of the sample (e.g., age, cognitive functioning) until later in the manuscript. "ADNI" and "HCP" mean different things to different people, so I'd suggest the authors provide basic cohort characteristics in the opening paragraph of the Results section so the reader can better understand the nature of data that were analyzed in this work.

This is a great suggestion. We have provided the cohort information in the opening of the Results section, and in a new Table 1 that summarizes study cohorts and their demographics. The ADNI dataset contains subjects whose ages fall in the range of 50 to 92, and their AD diagnosis can vary

from healthy control to dementia. On the other hand, the HCP Young Adults dataset features young healthy subjects who are 22-35 years old.

5. There is variability in imaging sequences and parameters across different phases of ADNI. This should be explicitly acknowledged, along with some details about how the authors accounted for this variability or not in their analytical approach.

Thank you for noting this important issue. We are unsure if the issue of phase-wise variability really applies to our chosen modality (T1-MRI). In all phases an isotropic 1mm resolution scan with full coverage of the brain was included, along with consistent standardization using phantoms across all sites. Further, each volumetric scan underwent our own consistent normalization to standardized atlas space via FSL. Hence our study design and data source may be considered safe from the vagaries of sequence variations that is certainly a real concern in other sequences and modalities used in ADNI. This is an important point we missed earlier and is now included in Discussion.

6. I think it would be fantastic if, as they say, the authors' algorithm can produce tissue segmentations "within seconds" that have comparable quality to those estimated more traditionally using FSL FAST. This should be backed up by empirical evidence and reported in the Results section, instead of just mentioning it in passing in the Introduction.

To further strengthen this claim, we are now including a benchmark of the processing time required by our UNet-based approach compared to standard bioimaging biomarkers and tools. See Result Section "Comparison with alternate models trained on tissue and regional volumetrics" and Table S4 (in Supplementary). In summary, the deep learning models achieved an average inference time of ~10s on a single target, while the FAST tool required 7 minute 34 seconds.

7. Table 1: Please provide the breakdown by AD dementia, MCI, and CN per set.

We have provided the breakdown of diagnostic groups in the new Table 1 (the previous SI Table 2).

8. The distinction between "AD" and "non-AD" mentioned on page 5 is ambiguous and needs to be more thoroughly explained in the Results section. It seems that the authors rather mean "AD dementia" vs. "non-AD dementia (i.e., CN or MCI)", but then saying that MCI represents a "non-clinical group" (p. 14) is simply incorrect.

We have adopted the suggested usage in revision: AD dementia and non-AD dementia. MCI are no longer referred to as a non-clinical group.

9. Also related to the diagnostic classification task, it's unclear what it means for the model to use both ADNI and HCP data as inputs, when the latter contained only young adults and no information about dx. Could the authors clarify? Ditto the analysis on predicting baseline cognition. This sounds contradictory to what the authors note in Online Methods, that the HCP YA data were used only for the purpose of augmenting the segmentation task.

HCP YA data were applied at the pre-training phase to serve the valuable purpose of augmenting sample size for the segmentation-only task. These brain MRIs from a younger group would benefit the model in reducing overfitting. The HCP YA data played an important role in enhancing the segmentation model by providing a set of adult brain scans without atrophy. This helps the segmentation task to be more generalizable and less error prone.

They were not included in other tasks like diagnosis or cognition task, because they are unrelated healthy subjects with no ADAS-Cog score assessment. Related text has been added to the “Datasets” subsection in Online Methods section.

10. Feature importance: Results shown on Fig. 4B deserve more investigation. I find it quite surprising and intriguing that cortical gray matter has little contribution to predicting cognition. If I’m reading this brain map correctly, it appears that the thalamus and some of the brainstem nuclei have the most significant contribution in predicting baseline cognition. This makes little sense in the context of clinical prognostication in AD just in terms of its pathological-degenerative cascade established in the field, provides very limited implications for possible interventions/clinical trials, and again makes me really wonder what the additional novel contribution this work has beyond what’s been already shown in the literature.

We agree with your assessment; the occlusion maps we had shown were indicative rather than thorough, and was based on a single representative subject’s MRI. You correctly note that this aspect needs a more careful and deeper exploration. In our revised analysis, we transitioned from generating occlusion maps based on a single representative subject to averaging occlusion maps across all AD subjects to enhance generalizability and robustness. The previous approach, relying on an individual subject, risked capturing subject-specific idiosyncrasies that may not reflect broader group-level trends. By averaging across subjects, the updated occlusion maps more accurately represent common, population-level predictive patterns and diminish subject-specific noise. Consequently, the results now appear significantly different due to the reduction of individual variability, highlighting consistent, generalizable features that influence the UNet’s predictive performance across the dataset. We also used a new, different software tool for visualization purposes, leveraging the open source Nilearn tool, which we believe serves our purpose better. Further, we added a new region-level visualization, this one using ENIGMA toolbox. The rationale for showing both types of maps is that they pick up different interesting features.

From the new occlusion maps, we conclude that the volumetric-to-surface projections particularly highlight the posterior parietal cortex, lateral and medial temporal regions (including hippocampus and parahippocampal gyrus), and frontal cortical regions as highly influential to cognitive score predictions. These regions align closely with established neuroanatomical correlates of cognitive impairment in Alzheimer’s Disease. However, we also found that the imaging hotspots were multifocal and widely dispersed throughout the brain. Yet, the fact that we were able to highlight hallmark structures with well known relevance in AD pathophysiology serves to further support the relevance of our modeling approach and suggests that we are picking up biologically plausible signals from raw MRI. We wish to thank the reviewer for requesting this analysis, which we believe has greatly enhanced the value of our study.

11. Data augmentation using image preprocessing: This section requires further clarification to be more accessible by the reader. Am I correct in that the authors manipulated some of the existing MRI data to generate synthetic images and treated them as if they had been acquired from new,

completely independent subjects? How many such cases were included, and would this pose a problem in terms of input dependency? Please also comment on what processing steps, if any, were implemented to input MRI data in general. Some MRI images have different resolution than 1 mm isotropic voxels – was resampling the only step that was involved? Any intensity normalization?

We applied elastic transformation to the MRI data simply as an augmentation tool in the training process alone, which is a common approach to make the DL models more robust. Note that above data augmentation strategy is limited to the training stage; when a trained model is applied to infer on test/validation data, only their true, unaltered MRI are used. These are fairly standard practices in the ML field. As far as resolution is concerned, we only used MRI data acquired at 1mm isotropic resolution, and all subjects underwent the exact same resampling to standardized space as described in Methods. No intensity normalization was performed. In each cross-validation training process, recall that our 9-fold cross validation strategy splits the full sample into 8/1/1 (training/validation/testing). Hence data augmentation was performed only on the first group, constituting 1560 subjects' MRI scan; the rest of the subjects in validation and testing groups were not affected by this step.

These points were added to the manuscript, both main and Supplement.

12. No code is provided in this submission.

R2 noted a concern regarding the availability of our code. We believe this is a misunderstanding, as we did provide access to the code through GitHub. We have reiterated and made the availability of the code more explicit in the revised manuscript. At acceptance we will be happy to further deploy the codebase, add more documentation and include a sample script to aid future researchers. We 100% believe in code sharing.

Reviewer #3

I enjoyed reviewing the manuscript and think it is interesting and novel. Predicting cognition from MRI is a relevant task. While the multimodality aspects of the paper are very intriguing, they are to some extent expected and prevent to fully judge the impact of the technological advancements the authors propose. The diagnosis and segmentation use cases seem of lower novelty and impact to me. I think the paper could use some work to foster a better understanding of details and the experimental set ups.

Thank you for the encouraging remarks. We are in alignment with your general assessment - this study was designed to be inherently multimodal but with an overriding emphasis on the cognition task - the key gap in the literature. While other tasks we successfully handled - diagnosis and segmentation - are less novel in comparison, we have hopefully convincingly argued that their inclusion serves a vital purpose: it enables our model to learn the implicit representations in the neuroimaging data in a manner that makes ALL prediction tasks more accurate, even if the user may prefer only one such task.

We hope also that our careful addressing of your detailed suggestions has improved the value of our study.

Major points:

Figure 1 is not clearly understandable. The authors talk about a ‘regression layer’ and the tree-based ‘regression module’ are those the same? Or is the regression layer just a linear layer to which the low-dimensional representations are fed? Which multitask predictions does the regression layer solve (the three shown on the right? What’s the dimensionality then?)? Why are the latent features fed into the ensemble model and more concrete features (voxel counts etc) to the boosting algorithm?

“Regression Layer” stands for the fully connected layers at the bottom of the UNet/MedNet that take latent variables as input and produce regression outcomes. “Tree-based Regression Module”, however, means the XGB module that is parallel with the Imaging Module. You have rightly pointed out the confusing aspects of the figure. We have redone this and hopefully reduced the confusion in Figure 1’s flowcharts as well as the text captions.

I don’t fully understand the value in having the model perform the three described tasks (segmentation, diagnosis, predicting cognition) in one approach. I understand that all of these can be of interest in different scenarios but seldomly they are needed all together. Is a model solely dedicated to predicting cognition worse? The authors refer to previous works but have not provided a head-to-head comparison for their datasets.

Yes, indeed, all 3 tasks are needed to be performed by components of the same model architecture, since our overall premise is that these different but related tasks serve to regularize each other and benefit from the implicit representations learned by the mapping from MRI to segmentation. To answer your question more concretely, yes, we have tried to point to evidence, both from the literature and from our own analysis, of this aspect. First, we showed (Model M3 in Table 2) that a CNN model trained to predict cognitive score from MRI does a poor job, due to the huge dimensionality mismatch between imaging data and scalar outcomes. A UNet model trained the same way, for only predicting cognition, performed better, but still not satisfactorily. The table also clearly indicates that the best performance comes from the multi-task setting where all features and outcomes are combined and trained together. Originally, we found this when we only worked on a certain subset of the ADNI data, and then we applied the multitask approach to the current more completed dataset. To provide the evidence that it’s still the case, we have added Supplementary Figure 1 to Supplementary Materials document with the scatter plot of our proposed UNet model with diagnosis predictions turned down, and the results showed that the testing R^2 was lower and MAE was higher than those from the full ensemble multitask model.

“While the segmentation task serves [...] to enable efficient learning in a data-scarce setting” there is no data shown in the manuscript to merit that conclusion. I see where the authors are coming from but without evidence, this remains speculative.

Thank you for pointing this out. In fact, segmentation is a substantial task for the structure of this model. In Table 2 we have clearly given the data to support this conclusion. Model numbered M3 in

the table corresponds to a CNN trained to predict cognition without an accompanying segmentation task. Its performance is significantly lower than those models trained using a segmentation task in addition to the cognition task (M4-M10). Thus, we hope we have supplied both reasoning-based and data-based support for this argument.

I would love to see a comparison not only against an XGB with demographics only but an XGB with FreeSurfer generated features (volumes, thickness). I think this would be a fairer comparison, since now the different access to information is biasing the model comparison and thus it's unclear whether the gain in performance is due to additional features/types of information or technological advancements made by the authors (plus, the data is readily available for ADNI and easy to access).

This is a valuable suggestion, also made by the editors. Please note that we had already provided some imaging benchmark results using a custom CNN architecture in the original submission (model M3 in Table 2). However, we agree with you and the editors that additional benchmarks involving volumetric data using established methods like Freesurfer or ANTS are needed to fully contextualize our approach and results. In the revision we are now adding a whole new section and new figure panels devoted to this task. See section "Alternate models trained on Freesurfer volumetrics" and new **Figure 5C-E**. These new benchmarks were directly compared to the proposed approach, and we were able to show that our approach outperforms carefully constructed models based on these features. We note also the somewhat utilitarian point about these computational pipelines being far more challenging and time consuming to implement, especially in clinical settings where such neuroimaging expertise may not be readily available. Our model in contrast does not require these pipelines and can work directly from a single MRI scan. These new data have hopefully added valuable context which was missing in the previous submission.

I would like to see further regression metrics besides the R2, to judge the clinical value of the predictions. The MAE and MAPE would be interesting to look at.

This is great advice. We have added the Mean-Absolute Error to all the modeling results in Table 2 in the main manuscript. We believe these metrics can help judge the clinical values better.

Fig 3A can only be interpreted when knowing how accurate the determined empirical alphas actually represent the trends in the data. I prefer the results shown in Fig 3 B and C. It would be interesting to have a Supplementary figure that gives an overview of the predicted alpha-parameterized trajectories and the empirical ones.

We appreciate your advice in creating an SI figure for presenting predicted trajectories against the real trajectories. This Figure has been added to SI Figure 3 along with the caption. Overall, the Pearson's R between predicted and actual trajectory calculated by UNet was 0.84, and was 0.85 from MedicalNet, both statistically significant with p-values smaller than 0.0001.

For Fig 4 A, y-axes should cover the same range. There is also a mismatch in dimensionalities as the image nets will feed significantly more features into the XGBs.

It would have also been interesting to see how much the latent representations add to the XGB over the explicit features calculated by the image nets.

We intentionally kept the y-axis ranges different in Fig. 4A to reflect the distinct nature of the comparisons. The left two bar charts specifically show the added value of the latent representations to the XGB models, on top of the explicit features extracted by the image networks. To account for the mismatch in dimensionality, we have produced additional analyses and figure panels that show the occlusion maps of imaging data directly. We have also shown in the last panel of this figure the latent representations captured during the cognition task, visualized on selected layers of the UNet model.

I would have liked to see a proper external validation on independent data (for example training on ADNI, testing on another cohort).

This is a valuable suggestion. We understand the importance of validating any machine learning model on independent and completely out-of-sample cohorts. To this end, we have conducted validation experiments using the data from the **Dallas Lifespan Dataset from the OpenNeuro platform**. This is an external MRI study on aging subjects unrelated to the ADNI study used in our manuscript, including 465 older subjects with MRI, of which 331 subjects have ADAS-Cog scores recorded in wave 2 and wave 3. OpenNeuro is a public, freely accessible and widely used platform; hence its proposed use as an independent cohort will not only strengthen the manuscript but also provide the wider community an additional resource.

This is a sufficiently large sample size that is more than adequate to support the analysis from the main study. Since the Dallas cohort does not include mature Alzheimer patients, it has additionally served an instructive purpose to assess the applicability of our models trained on the AD spectrum to this new cohort. Please note, although there are other studies with larger samples, they typically lack a comprehensive ADAS-Cog battery, which constituted our main outcome measure.

The results of this independent validation were quite remarkable. Our ensemble UNet on the DLBS dataset achieved out-of-sample testing R^2 score of 0.63, and an MAE of 1.69. These results can be found in the newly added Figure 5. *The fact that the model, trained only on the ADNI study, was able to accurately predict cognitive scores in a completely different dataset without undergoing any additional re-training, supports the generalizability and robustness of our model.* We thank the reviewer for this suggestion, which has made our study considerably more impactful.

“Both scale parameters σ and γ are estimated directly from empirical data at the pre-processing stage, without further learning.”

- The parameters need to be calculated inside the CV. Otherwise you will run into overfitting through data leakage

You are correct. In the exploration process, we first found that the empirical cognitive scores fall into a Gamma distribution and then trained an experimental XGB model using the empirical numbers by analyzing the whole dataset to testify the custom loss function. Then, at the actual model training phase, for each cross validation split, the parameters σ and γ were then calculated only within this training cohort, which would not cause any data leakage. We have updated this part of the manuscript.

- How would this transfer to other cognitive outcomes that may not follow the same distribution (i.e., CDR and CDR Sum of Boxes scores, the latter of which is often used in clinical trials for AD)?

This is an interesting question that we have not fully explored. For now, we offer the reasons we chose not to use CDR for our purposes. Limitations of the CDR include its length of administration, reliance on clinical judgment and collateral source information, and relative insensitivity as a measure of change in interventional studies. (*Utility of the Clinical Dementia Rating in Asian Populations*). CDR is also only a number in the range of (0, 0.5, 1, 2, 3), which are ordinal measures rather than continuous ones, due to which it does not appropriately constitute a regression task. Now, whether and how much the training is transferable to CDR: this will require a more in-depth study than current scope allows. The literature does not indicate an overly strong correlation between these metrics, since the component subscores are frequently measuring very different things. Nonetheless, this question can and should be amenable to future quantitative analysis.

It is not clear how class imbalances were attributed for when presenting the accuracy of the diagnosis prediction.

To this end, we have added a new confusion matrix in Figure 2C. Suggested by the results, we can safely conclude that the class imbalances are not hurting the accuracy.

“Accuracy increased significantly to 94.56% when we combined the UNet and XGB models into a multi-task approach, leveraging both ADNI + HCP inputs and demographics”, does this imply that the results were only produced on one of these data sources? If so, the performance gain is not necessarily due to the technical contributions of the authors.

The results were only produced on ADNI data, because ADNI has actual diagnostics in its public files. HCP subjects are healthy control only so it would be inappropriate to predict diagnosis on them. We have amended this sentence slightly to remove the unintended implication that the diagnosis accuracy improvement is mainly due to the use of HCP data. In fact, the improvement is due to the joint use of imaging and demographics, in a multi-task and ensemble setting. A closer look at Table 2 would confirm this conclusion. We also note that while we are reporting diagnosis prediction performance, it is not intended as a novel, standalone outcome of our technique.

“Data augmentation using image pre-processing”

The authors should provide more specific information in the Supplementary Material.

We have provided more training and augmentation details in Supplementary Information document. We also edited the Data Augmentation subsection in main Methods section for clarity. In brief, the data augmentation process includes elastic transformation of the training images. It generates displacement vectors for all pixels based on random offsets to enhance the generalizability of the models. See SI Section “Supplementary Methods - Data Augmentation”.

From the methods it is not always clear based on which performance decisions about model architecture were made (e.g., deciding for the XGB module and going for the Resnet50 instead of 10). Such decisions should be made solely based on the CV results.

We have expanded on the description of model choices. We added a new subsection entitled: “Model Selection Decisions” in Supplementary Materials. We quote:

Supplementary Table 2 shows the preliminary results from the exploration phase of our study, where we simply split the data into 9:1 training and validation sets, and compared multiple modeling options’ performance on the validation set, to instruct our selection of machine learning models. Due to the huge diversity of model options and to economize on overall computational burden, we found it sufficient to assess the suitability of model selections from a single run of training/validation. From the top 4 rows of the table, we can safely conclude that the XGB architecture outperformed the Random Forest and Support Vector Regressor on the baseline cognition task, by achieving R2 scores over 0.2. The bottom 4 rows, on the other hand, showed that the “3DResNet50” Model in MedicalNet’s codebase was superior to the “3DResNet10” Model, compared on both ADNI MRI inputs alone, as well as the ADNI+HCP imaging inputs.

In fact, not all the modeling decisions are made based on cross validations. At the exploration phase of this study, we made the decisions of XGB over other conventional ML techniques through a single validation comparison. (See SI Table 2) Therefore, we have selected the ResNet50 and XGB modules based on the experimental results. We have made this clear in the Supplementary Section “Model Selection Decisions”.

Are the XGB and DNN trained in v? The L_DNN and L_XGB makes it seem they are not. I would be interested to see whether an end-to-end training makes the system better (as image net-derived features are specifically catered to the cognitive prediction task). Or is L_Cog in L_DNN derived from the XGB output (I guess its from the ‘regression layer’ mentioned in Fig 1)? The entire set-up could be described a bit more clearly.

L_Cog is the loss function designed for cognition loss, when it’s applied in L_DNN (now renamed to L_Im), the inputs are derived from the regression layers of the Imaging model. The XGB model and the DNN model were trained in parallel. We realize that prior text describing the various loss functions was not as clear as we intended. The revised Methods now has new Loss function subsection and Ensemble subsection that have been rewritten and refined based on this advice.

Minor:

- Regarding accurate language, it seems like the authors here are not targeting AD as it is defined nowadays but the clinical syndrome of AD, being AD dementia.

We are targeting the AD dementia using the diagnostic categories provided by ADNI dataset and other datasets. Based on another reviewer’s comment we have clarified also that MCI and AD are purely syndromic categories, and that only the AD group may be considered to have AD-dementia (beginning para of Results).

- I think it would be good to mention that the “3D ResNet50 architecture” mentioned in the introduction is later referred to as MedicalNet.

Thank you for pointing this out. We have reorganized the mentioning of certain models for them to occur in a more accurate order.

- In the legend of Table 1 it is written that a 10-fold CV was conducted. In the Segmentation section it says 5-fold

Apologies for this clerical error. In fact, we implemented 9-fold cross validation throughout. Text has been updated everywhere to accurately reflect the latest approach we applied when we collected the final results presented in the paper.

- Talking about “excellent” results reads a bit like confirmation bias to me. I would prefer no judgement about the results in the Results section and letting the numbers speak for themselves.

We have removed the subjective descriptions of the model results. We’ve replaced and removed most subjective words such as “excellent” or “decent”.

- Typo: “r4eently”

- “Tencent” spelling is inconsistent

These have been fixed. Thanks for pointing them out.

Dear Editor,

Thank you for arranging the second round review of our manuscript. We were thrilled by the enthusiastic and positive comments by the reviewers. We consider this version to constitute quite minor revisions, which we have performed. We have fixed several minor issues identified by reviewers, and have performed new analysis and tabulated results as requested by reviewers. The manuscript has consequently improved. We are uploading herewith a clean as well as a highlighted version of the manuscript.

The following contains a line-by-line response to the reviewers' advice. We are confident that these revisions will have fully satisfied the reviewers.

Best,

Daren Ma, Yang Yang and Ashish Raj, on behalf of all co-authors

Reviewers' Comments:

Reviewer #1 (Remarks to the Author):

- Overall, the authors have done a very good job on responding to the reviewers' comments, and the manuscript is definitely improved, particularly in terms of clarity and language.

We are grateful for your thoughtful assessment. It is encouraging to know that our efforts in addressing the reviewers' comments have improved the manuscript vastly, based on the constructive feedback of all reviewers.

- The description of the ensemble approach is improved, though I think needs a bit more detail. Firstly, I haven't come across an ' R^2 ensemble' approach before. I assume this serves to weight the base models by the variance explained by the Imaging module and non-imaging module. So, if the modules had similar R^2 , then actually this ensemble would have limited effect? I think it would be useful to clarify exactly what the R^2 values from the cross-validation were for each of the modules, so that the effect of the ensemble is more transparent. This is an important point because one possibility for the improvements after the ensemble is simply that the network was trained for more epochs.

We thank the reviewer for this feedback on one of our novelties in our algorithm. We have utilized the 'R² ensemble' approach in the pretraining steps in order to let the ensemble benefit from the pre-trained performance of its constituent modules. If the modules had similar R², the resulting loss function would be directed to treat both modules equally during the full retraining phase. If one or the other module had exceptionally high R², then that module will be dominantly favored in the ensemble approach. This is a designed behavior as we rely on the R² scores alone to decide the relative weights without excessive tweaking. Although we are aware that other more sophisticated ensemble techniques are available, here we were focused on the simplest possible strategy, which also helps in reducing the risk of overfitting. We have added this to revised manuscript (Methods).

Regarding the suggestion about reporting the R² of each module separately, we have already reported that in Table 2, which shows the performance achievable by each module separately (M2 (XGB) R²: 0.31, M7 (UNet) R²: 0.72). From the respective R² of the cross-validated performance of the two models being ensembled, M2 and M7 (see Table 2), it is observed that the ensemble approach approximately weighed the XGB portion by 30%, and the UNet portion by 70%. This has been added to the manuscript (Results). Regarding the effect of number of epochs, we have reported in Supplementary Figure S2a that after about 46 epochs the UNet model, for example, stops improving. Hence we do not believe that the ensemble performance can be explained simply by increasing the number of epochs.

- Building on this, generally there is a lack of mention of the numbers of epochs used for training, and whether over-fitting was evaluated. Clearly some overfitting occurred, as the performance in the additional DLBS data explains considerably less variance than on ADNI.

The epochs of training and their effects have been mentioned in the Supplementary Figure S2. This figure clearly shows that after about 46 epochs the UNet model stops improving and the MedicalNet stops improving after 44 epochs. Evidence for over-fitting is also available on this plot, since it shows that while training loss continues to improve for higher epochs, the validation loss hits a minimum at 40-50 epochs. This was added to revised manuscript's Supplementary section.

Indeed, the model performance in ADNI is superior to DLBS, however, in this comparison of modeling results, we attribute the performance difference primarily to domain shift, as ADNI and DLBS differ in scanner distribution, demographic composition, and acquisition parameters. For instance, the ADAS-Cog-11 scores in DLBS range from 0 to 19.3, with an

average of 5.2, however, the average ADAS-Cog-11 of ADNI cohort is near 10, and the distribution is 0-42.7 (these statistics can be found in Table 1). This was added to revised manuscript's Results section.

- In fact, considering the DLBS dataset, it would have been good to test all their models (i.e., M1-M10) on the DLBS data, not just M9. This would help quantify whether the top-line ensemble performance in the ADNI is due to ensemble over-fitting or whether the performance improvement is maintained out-of-sample.

We understand the concerns of over-fitting raised by the reviewer. To further address this issue, we have listed all the DLBS testing results in a new Supplementary Table S6. We believe the new results show that the performance is mainly caused by the heterogeneity of the datasets themselves, instead of the overfitting phenomenon. We have put a screenshot for that Table below.

Table 6 The modeling testing performance for DLBS cohort set. The relative performance of models M1 to M10 is similar in the DLBS cohort in comparison to the ADNI cohort. Note that the range of ADAS-Cog-11 in DLBS is narrower than that from ADAS, therefore the MAE for DLBS testing are generally lower.

Model	ADNI Testing R^2	ADNI Testing MAE	DLBS Testing R^2	DLBS Testing MAE
M1 XGB (Standard MSE)	0.24	6.69	0.10	5.10
M2 XGB (Gamma Loss)	0.27	6.45	0.12	4.24
M3 Single-Task CNN	0.42	4.18	0.25	3.49
M4 Multitask UNet	0.60	4.92	0.31	3.15
M5 UNet	0.66	4.31	0.39	2.78
M6 MedicalNet	0.58	4.48	0.33	3.11
M7 UNet	0.68	3.97	0.45	2.36
M8 MedicalNet	0.70	2.82	0.44	2.57
M9 Ensemble(UNet +XGB)	0.82	2.48	0.63	1.69
M10 Ensemble (MedicalNet +XGB)	0.80	2.29	0.60	1.88

Reviewer #2 (Remarks to the Author):

The authors have thoroughly considered and addressed most of the concerns I noted in my initial review of this work. Regarding Comment #5, the authors' statement regarding parameter variability across ADNI phases is incorrect. It wasn't until ADNI3 when MPRAGE was acquired using 1-mm isotropic voxels. ADNI Go/ADNI2 had slice thickness of 1.2 mm. Perhaps the authors are confusing "phases" with "sites". In any case, for full transparency, I still suggest the authors report which ADNI phases the current subject sample represented. Even if the authors chose not to do so, their current statement about all ADNI phases associated with 1-mm iso MPRAGE is inaccurate and I'd advise them to drop it.

We appreciate this thorough and complete advice from the reviewer. We agree and have dropped the claim that all the T1-weighted scans are of 1 mm slice thickness. Moreover, we report that the current subject sample represented ADNI 1, 2, GO, and 3, and report the actual voxel resolutions of each. We have updated the text in affected paragraph as such:

In this study, we included baseline scans of healthy participants from the ADNI-1, ADNI-2, ADNI-Go, and ADNI-3 cohorts. All scans were acquired using an MPRAGE sequence with parameters of TR = 2300/2400–3000 ms, TE ≈ 3 ms, TI = 900–1000 ms, and flip angle = 8°–9°. The images were obtained across multiple sites using scanners from three different manufacturers (GE, Siemens, and Philips). The spatial resolution of ADNI-1 images was 0.9375 mm × 0.9375 mm × 1.2 mm, while that of ADNI-2/Go images was 1 mm × 1 mm × 1.2 mm (Gunter et al., 2009). For the ADNI-3 cohort, the subjects were acquired at 3 T across multiple vendors (GE, Siemens, Philips) using harmonized protocols, with T1-weighted MPRAGE volumes obtained at a resolution of 1 × 1 × 1 mm³ (Gunter et al., 2017). We did not attempt harmonization between phases for ADNI.

Reviewer #1 (Remarks to the Author) **based on Reviewer #3' report** in the previous round:

I've gone through the rebuttal to reviewer #3 and I think the authors have done a good job of addressing their concerns.

However, I noticed something else that requires clarification. The authors now introduce a confusion matrix into figure 2. The values in the matrix seemed a bit surprising (particularly the number of false negatives for model M10), so I calculated the accuracy and other measures myself from the numbers they report here. Unfortunately, the results don't match up with the results in Table 2, or Figure 2C and 2D. Instead, the confusion matrix results give accuracies of M9=0.8876 and M10=0.8203, which is considerably lower.

Perhaps this is just a typo or I'm missing something, but this potentially questions the results throughout. Hopefully, it's just simple mistake that can be rectified, but needs clarification before publication, and perhaps motivates the reporting of confusion matrices for all the results (in the supplementary material).

We thank the reviewer for noting this small but important discrepancy. This was entirely our fault, due to a clerical error that caused an error in generating the confusion matrix for Figure 2C. Now, we have updated the confusion matrix through combining the validation sets from each cross-validated split and the testing, with the full cohort of all 1950 samples

from ADNI. The results now reflect the genuine accuracy reported in the plot. The new confusion matrix for M9 gives an accuracy of $(1450+394) / 1950 = 94.56\%$, and M10 of $(1433+393) / 1950 = 93.64\%$.

Further, we thank you for the suggestion to include a confusion matrix for all reported models. We have added the cross-validated confusion matrices for all the models on the diagnosis task, as a new Supplementary Table 7.

Response Document

Dear Editor,

We have gone through the reviewer's last remaining critique very carefully. They have raised constructive comments regarding the inconsistent manner in which we have reported our cross-validation results and Diagnosis Accuracy. We thank the reviewer for raising this issue, which is indeed important for clarity of our modeling performance, and we have the responsibility to further clarify it in the updated manuscript. These changes involve showing additional and more detailed information rather than any errors in calculation. With this we believe we have addressed the reviewer's remaining concerns.

Appended: The reviewer's comment and our response:

- Good to see that they've corrected the confusion matrix, though this has raised another important question. Table 1 and Table 2 say the ADNI test set $n=195$ and the results in Table 2 are from the 'set aside testing', yet the results in Table 2 are exactly the same as the confusion matrix, where the $n=1950$ (i.e., M9 accuracy in Table 2 is 94.56%, which is the same as the updated confusion matrix in Figure 2 and Supplementary Table 7). It's not impossible to get identical results to the 2 decimal place when evaluating on $n=195$ vs. $n=1950$, but seems unlikely. Can the authors confirm what the sample size used in Table 2 was?
- Either way, it's unclear how the predictions were generated for the $n=1950$ confusion matrix in Figure 2/Supplementary Table 7. On page 12, they describe the validation scheme as either 9:1 or 8:1:1. Both imply that the results are only being evaluated on the 'set-aside' test set (i.e., $n=195$), not on all the ADNI data ($n=1950$). This section needs updating to explain how they generated predictions that populate the updated confusion matrix and supplementary table 7. I realise they used cross-validation within the 90% training data, but that shouldn't really be reported as 8:1:1. More importantly, it also means that 90% of the predictions were generated within the training cross-validation while the remaining 10% were truly set side. However, the results in Figure 2/Supplementary Table 7 are reported all together, ignoring this distinction between how the training and test were split, which is not really appropriate. Moreover, it does not solve the issue of the identical results in the confusion matrix and Table 2, which purportedly come from different (overlapping) test sets.
- I think it is essential that this issue is clarified before publication. I hope this is not the case, but it's possible that there have been some errors in how the results were generated which could have caused data leakage and inflation of performance.

We agree that the issue requires some clarity and further detail. In our previous version we had omitted some of these details in order to economize on the length of text and tables. In this revision these details have been carefully assembled.

First, we want to assure that all the numbers provided are from validation and testing sets, and there was no data leakage. The number of samples in the previous Table 2 was 1950, combining of all validated samples in the cross-validation ($n=1755$), plus those in the set-aside testing set ($n=195$), which was never exposed during the cross-validation steps. We realize this

presentation left the unintended impression that we were reporting set-aside testing result for the diagnosis task in Table 2.

Therefore, in the updated manuscript, we have renamed Table 2 to “Main Task Modeling Results” and have reported both CV and set-aside testing scores for all three tasks - Diagnosis Accuracy, Segmentation Dice Scores and cognition prediction. In this Revision, based on your concern we have carefully reported three results for the diagnosis task – for the full 1950 dataset; for the 1755 training+validation dataset; and for the 195 set-aside testing dataset.

We have added Supplementary Tables 8 and 9, and Supplementary Figure 4 to display the actual testing accuracy of the selected models, as well as the complete results of confusion matrices. We have chosen to keep in Figure 2 the ROC curves and confusion matrix arising from the full dataset of 1950 subjects, however, all three sets are provided as shown above.

We also restructured the subsection “Cross-validation and set-aside data splits” in the Methods section, fixing detailed wording used to describe our cross-validation method. Now the text clearly says that:

For this study, we collected 1950 ADNI subjects in total, and performed a train-test split at a ratio of 9:1. This 10% testing set (n=195) is set-aside and never used in the training or validating processes, until we report the testing performance of each model. On the remaining 90% of samples (n=1755), we performed 9-fold cross validation for all the models. It is worth pointing out that when we report the main results or decide model selections, we always refer to the validation sets in the 9-fold CV process, and never relied on the training scores. In order to eliminate potential data leakage, that is, the possibility that the deep learning model "guesses" the outcomes from patient-specific data instead of learning the true, broad underlying patterns, we made sure that the patient ID information was preserved across all stages of data cleaning in the curated dataset, but then they were dropped before any model fitting, which guaranteed no overlapping subjects in the splitting. Further, we applied the same random seed in performing the train-validation splitting, so that the cross-validation results were reliable and comparable across different models.

Further, we have made sure that the text regarding cross-validation is consistent for all tasks and everywhere in the text, accurately reflecting our 9-fold CV approach, with 195 samples of set-aside testing set. Please see the highlighted text in the updated manuscript.

We thank the reviewer again for their rigorous comments on the diagnosis results of our paper. We have now fixed the issues raised above and believe it will address the concerns of performance inflation.

Sincerely,
Daren Ma, Ashish Raj and Yang Yang
University of California San Francisco
{daren.ma, ashish.raj, yang.yang}@ucsf.edu

Reviewers' Comments:

Reviewer #1 (Remarks to the Author):

Good to see that the authors have updated the manuscript a bit, but I still don't think it's appropriate to merge results from CV and held-out testing. CV is known to inflate performance relative to hold-out, so they really should be reported separately. Also, Figure 2 remains a bit confusing with some of the results presented just on the held-out (e.g., panel G) with others on the merged CV and held-out test data (e.g., panel D). While it explains this in the legend, I don't see the motivation for doing it.

We are grateful for this attention to detail. We agree that the modeling performance will be more accurately highlighted if we display the set-aside testing and cross-validated scores separately. Therefore, we have modified Figure 2 C and D in such a way that now it only contains set aside (testing) results in every subplot. We have moved the full dataset results (training, cross validation and testing performance) to SI Figure 4, along with the corresponding AUC results, to more comprehensively illustrate the prediction power of the diagnosis tasks. We believe this fully resolves the reviewer's last remaining concern.